# Topographic mapping of the glioblastoma proteome reveals a triple-axis model of intra-tumoral heterogeneity

K. H. Brian Lam [1], Alberto J. Leon[2], Weili Hui[2], Sandy Che-Eun Lee [2,3], Ihor Batruch[4], Kevin Faust [2,5], Almos Klekner[6], Gábor Hutóczki[6], Marianne Koritzinsky[2,3,7,8], Maxime Richer[9,10], Ugljesa Djuric[1,2,11] & Phedias Diamandis [1,2,3,11✉]

Glioblastoma is an aggressive form of brain cancer with well-established patterns of intra-tumoral heterogeneity implicated in treatment resistance and progression. While regional and single cell transcriptomic variations of glioblastoma have been recently resolved, downstream phenotype-level proteomic programs have yet to be assigned across glio-blastoma's hallmark histomorphologic niches. Here, we leverage mass spectrometry to spatially align abundance levels of 4,794 proteins to distinct histologic patterns across 20 patients and propose diverse molecular programs operational within these regional tumor compartments. Using machine learning, we overlay concordant transcriptional information, and define two distinct proteogenomic programs, MYC- and KRAS-axis hereon, that coop-erate with hypoxia to produce a tri-dimensional model of intra-tumoral heterogeneity. Moreover, we highlight differential drug sensitivities and relative chemoresistance in glio-blastoma cell lines with enhanced KRAS programs. Importantly, these pharmacological dif-ferences are less pronounced in transcriptional glioblastoma subgroups suggesting that this model may provide insights for targeting heterogeneity and overcoming therapy resistance.

[1] Department of Laboratory Medicine and Pathobiology, University of Toronto, Toronto, Ontario M5S 1A8, Canada. [2] Princess Margaret Cancer Center, University Health Network, Toronto, Ontario610 University Avenue, M5G 2C1, Canada. [3] Institute of Medical Science, University of Toronto, Toronto, Ontario#2374-1 King's College Circle, M5S 1A8, Canada. [4] Department of Pathology and Laboratory Medicine, Mount Sinai Hospital, Toronto, Ontario M5G 1×5, Canada. [5] Department of Computer Science, University of Toronto, 40 St.George Street, Toronto, Ontario M5S 2E4, Canada. [6] Department of Neurosurgery, Faculty of Medicine, University of Debrecen, 4032 Debrecen, Hungary. [7] Department of Radiation Oncology, University of Toronto, Toronto, Ontario#504-149 College Street, M5T1P5, Canada. [8] Department of Medical Biophysics, University of Toronto, Toronto, Ontario M5S 1A8, Canada. [9] Department of Pathology, Centre Hospitalier Universitaire de Sherbrooke, 3001, 12e avenue Nord, Sherbrooke, QC J1H 5N4, Canada. [10] Axe neurosciences du Centre de recherche du Centre hospitalier universitaire (CHU) de Québec–Université Laval et Département de biologie moléculaire, biochimie et pathologie de l'Université Laval, Québec, QC G1V 4G2, Canada. [11] Laboratory Medicine Program, University Health Network, 200 Elizabeth Street, Toronto, ON, Toronto, Ontario M5G 2C4, Canada. ✉email: p.diamandis@mail.utoronto.ca

Glioblastoma (GBM) is the most common and aggressive form of brain cancer with a dismal prognosis of approximately 15 months survival despite multi-modal therapy[1,2]. Classically, GBM has been defined by hallmark histomorphological "multiform" features including cellular (CT) and infiltrating tumor (IT) areas, regions of microvascular proliferation (MVP), and hypoxia where tumor cells palisade around necrosis (PAN). These diverse histomorphologic niches, found in varying amounts across cases, create a heterogenous tumor microenvironment (TME)[3] with molecular attributes that contribute to tumor evolution and treatment resistance[4–6]. Previous large-scale transcriptome profiling efforts such as The Cancer Genome Atlas (TCGA)[7,8], Repository for Molecular Brain Neoplasia Data (REMBRANDT)[9], the Human Protein Atlas (HPA)[10] and, most recently, CPTAC initiative[11] have largely focused on defining overall inter-patient molecular differences of bulk tumor specimens. Using multi-modal genetic sequencing techniques, the two former initiatives defined genomic alterations in specific GBM sub-types including IDH1/2, ATRX, and p53 mutations particularly enriched in younger patients with more favorable outcomes. Conversely, IDH-wildtype GBMs frequently presented with alterations in the phosphatase and tensin homolog (PTEN), the telomerase reverse transcriptase (TERT) promotor, and regions of chromosome 7 harbouring the epidermal growth factor receptor (EGFR) gene. More recently, the HPA used tissue microarrays and systematic immunohistochemical (IHC) staining to non-quantitatively profile protein patterns in relatively small tissue cores of multiple GBM specimens. While these initiatives have created valuable resources and have transformed our understanding of GBM, they were not designed or intended to quantitate potential intra-tumoral molecular landscapes of GBM's heterogeneity that are now implicated in resistance models of cancer.

Emerging tools and approaches have recently begun to comprehensively characterize cell-to-cell and niche-specific genetic differences within GBM. Single-cell genomic profiling studies have highlighted multiple dynamic transcriptional states of cancer cells within GBM that may help explain resistance to current therapies[12,13]. Complementary approaches by the Ivy Glioblastoma Atlas Project (Ivy GAP)[3] used laser capture microdissection (LCM) to regionally isolate and define transcriptional profiles across hallmark histomorphologic niches. Importantly, distinct transcriptionally-defined cellular states appear to, at least in part, be regionally segregated within Ivy GAP's anatomical niches suggesting strong contributions of the microenvironment to GBM's intra-tumoral molecular heterogeneity[14,15]. Together, this has revitalized histomorphologic-driven approaches to tissue profiling designed to resolve consistent and targetable niche-specific molecular dependencies and vulnerabilities[16,17]. These include the downstream effects of hypoxia-inducible factor (HIF) pathway and tumor necrosis factor (TNF) signalling in hypoxic niches (PAN)[18–20], PI3K and angiogenic pathways in tumoral vessels (MVP)[21,22], and differentiation and proliferation programs in cellular tumor regions (CT)[23–25]. While the Ivy GAP has already proven useful in developing hypotheses related to GBM pathogenesis and treatment, accumulating proteogenomic comparisons in many cancer types have now shown that protein levels cannot be reliably inferred from mRNA measurements ($r = \sim 0.35$)[26–29]. This is further demonstrated in recently published proteogenomic interrogations of glioblastoma where there are both correlative and divergent protein-to-RNA level relationships[11]. Therefore, a spatial proteomic resource would complement these existing resources, and help further refine our phenotypic understanding of niche-specific molecular heterogeneity of GBM,

Here, we developed a liquid chromatography tandem mass spectrometry (LC-MS/MS)-based proteomic atlas that aligns abundance levels of 4794 proteins to hallmark histomorphologic niches (accessible at: https://www.brainproteinatlas.org/dash/apps/GPA). This effort highlighted a substantial number of regionally enriched proteins and distinct molecular programs compared to the overall bulk tumor signature. In addition to traditional supervised approaches to gene set enrichment analysis (GSEA), in which differential biological programs are defined by different human-annotated sample labels, we also perform a region-agnostic approach to define heterogeneity in GBM tumor regions. By carrying out single-sample gene set enrichment analysis (ssGSEA) and using XGBoost machine learning regression models to independently reveal the functional status of each sample, we uncover a regionally-variable MYC-KRAS molecular axis that cooperates with hypoxia to generate intra-tumoral heterogeneity. Using multiple independent datasets, we show that tumors over-represented along the KRAS axis show a clinical aggressive and invasive phenotype, while GBM cells and tumors showing a MYC-bias largely display proliferative programs. Lastly, using computational pharmacogenomic models and drug screens of glioma stem-like cells, in both normoxic and hypoxic culture conditions, we highlight important drug sensitivity differences across the various axes and a tendency for KRAS-like cell lines to be relatively resistant to pharmacological inhibition. Importantly, we show that pharmacological sensitivities were more prominent along our proposed protein-defined axes than those defined by the classic transcriptional subgroups of GBM[30]. Together, the developed atlas and defined tri-dimensional MYC-KRAS-Hypoxia axis model provides important insights and tools to help guide the design of drug combination strategies that could potentially target an overall larger fraction of GBM's heterogenous biology.

## Results

**Proteomic profiles of histomorphological niches highlight regionally unique biological processes.** To address if proteomic patterns vary across GBM's histomorphologic niches, we assembled a cohort of 20 well-sampled formalin-fixed paraffin-embedded (FFPE) IDH-wildtype primary GBMs by reviewing a larger array of representative cases from our institute to normalize for sample-to-sample morphologic variations and tissue purity (Supplementary Data File 1). For each case, anatomical niches were defined by consensus annotations by two board-certified neuropathologists (P.D. and M.R.) and laser capture microscopy (LCM) was used to excise regions of infiltrating tumor (IT), cellular tumor (CT), palisading cells around necrosis (PAN) and microvascular proliferations (MVP). Adjacent histologically normal brain tissue near the tumor's leading edge (LE) was also isolated (Fig. 1). In total, we isolated 86 LCM samples for analysis by LC-MS/MS resulting in quantification of a total of 4794 proteins across the entire sample cohort (Supplementary Fig. 1).

Multi-dimensional scaling through principal component analysis (PCA) demonstrated that sample clustering was largely driven by proteomic patterns of anatomical niches rather than inter-patient differences (Fig. 2a–c). As a comparison, we included a number of non-microdissected whole tissue samples that exhibited high variability on the PCA plot, a large interquartile range and an overall low median Pearson correlation (~0.71) of quantified proteins (Fig. 2b–d). Together, this highlights potential regional biases that could arise by performing bulk tissue profiling and strong niche-specific contributions to GBM protein patterns (Supplementary Figs. 2–25). Overall,

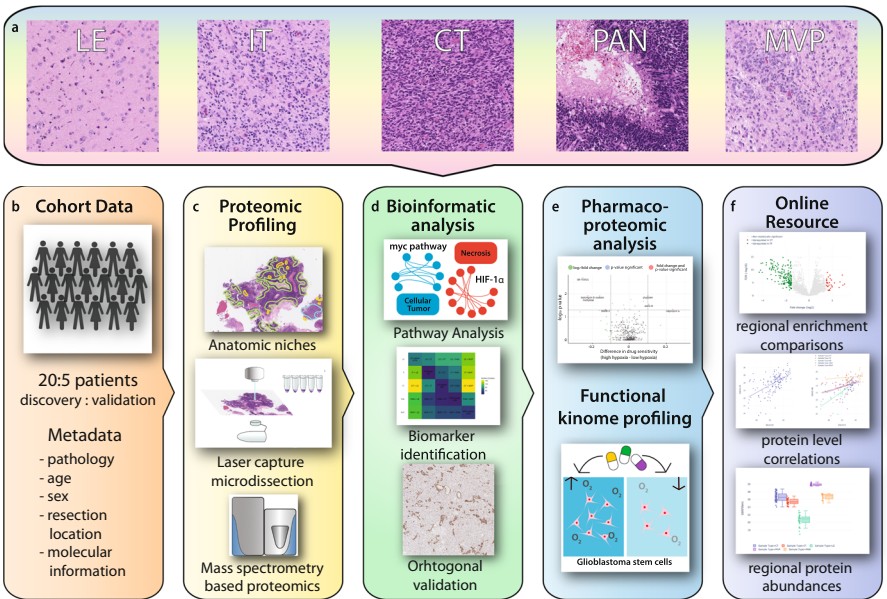

**Fig. 1 Schematic overview detailing data generation, analysis and presentation. a** Hematoxylin and Eosin (H&E) images detailing the anatomical niches within GBM: leading edge (LE), infiltrating tumor (IT), cellular tumor (CT), palisading cells around necrosis (PAN), and microvascular proliferations (MVP). **b** Collection of clinical data for a cohort of 20 GBMs. **c** FFPE tissue was sectioned and stained with H&E to identify histomorphological features. These were then excised using LCM and prepared for LC-MS/MS-based proteomics. **d** Bioinformatic pipeline including multi-dimensional scaling, differential expression matrix analysis, and gene set enrichment analysis (GSEA). **e** Selected region-specific biomarkers were validated within an external cohort of patients by immunohistochemistry (IHC). Clinical significance of regions specific markers was approximated using stratified RNA levels from TCGA. **f** Data was deposited in an accessible online data portal to allow for interactive and real-time exploration and comparison of proteomic profiles within GBM: (https://www.brainproteinatlas.org/dash/apps/GPA).

tumor-derived samples (IT, CT, PAN) of the same histomorphologic identity across patients strongly clustered together (Fig. 2c) with small interquartile ranges and a median Pearson correlation of ~0.83 (Supplementary Fig. 26). This ability to resolve robust proteomic concordance within specific GBM niches supports a strong TME component to previously described intra-tumoral molecular heterogeneities seen in GBM; findings consistent with transcriptomic data from Ivy GAP.

Hierarchical clustering of Pearson correlations across profiled samples further supported our PCA analysis with the largest similarities occurring within anatomical niches rather than patients (Fig. 2d). Supervised GSEA highlighted a number of pathways enriched at the protein level within the histomorphologic niches (Fig. 2e). Adjacent normal neural tissue at the tumor's LE was enriched for Gene Ontology terms related to neuronal systems while the IT region was associated with neuronal systems as well as stem cell-related pathways. This latter enrichment is consistent with recent studies implicating cancer stem-like cells in driving infiltration and distant spread within the nervous system[24,31]. CT regions were associated with differentiation and growth, whereas PAN was associated with stress, hypoxia, and immune responses. MVP showed enrichment of programs involved in angiogenesis, immune regulation, and response to wounding. Together this supports that the GBM proteome can be regionally deconstructed into niche-specific signatures that can help reduce complexity and offer more refined biological insights into regional phenotypic programs.

**Proteogenomic comparisons and validation of niche-specific markers**. To validate a selection of niche-specific markers and targets nominated by our approach, we performed proteogenomic comparisons using Spearman rank correlations with transcriptional profiles from Ivy GAP (Fig. 3a). The highest correlations of ~0.4 were found between similar micro-dissected regions, a value

congruent to other recent proteogenomic comparisons[26–29]. We reasoned that candidates with high regional enrichments, both by RNA and protein, would represent attractive markers for validation. Towards this, we used a differential protein abundance cut-off of $p = 0.01$ to highlight a total of 1314 regionally enriched proteins (Fig. 3b and Supplementary Data File 2). Comparison with regionally enriched genes from Ivy GAP revealed that 494 of these proteins also had concordant elevation of their mRNA precursors (Fig. 3c and Supplementary Data File 3). These candidates, mutually elevated at both the RNA and protein levels in specific GBM niches, from independent studies, reveal the important non-random regional contribution to GBM's molecular heterogeneity. Conversely, we identified a subset of 820 markers that are defined as elevated only at the protein level and a subset of 209 genes that were detected to be only elevated at the RNA level (Supplementary Data File 3). These contrasting patterns, conversely offer an important avenue to understand pathway-specific processes regulated through post-transcriptional mechanisms.

We validated the regionally restricted patterns of representative candidates using orthogonal immunohistochemical approaches and independent patient samples (Fig. 3). This included confirmed strong and relatively selective staining of A-kinase anchoring protein 12 (AKAP12), a scaffold protein previously shown to regulate hypoxia-mediated growth and invasion in melanoma[32], in PAN regions (Fig. 3d–f). Such proteins and pathways could offer potential insights into hypoxia-related dependencies within this GBM niche. Similarly, in addition to numerous mesenchymal-specific proteins, we also confirmed enrichment of the immunoglobulin CD276 (B7-H3) in MVP regions (Fig. 3g–h and Supplementary Fig. 27). This non-canonical immune checkpoint protein is known to suppress T-cells response in other cancer types[33]. Importantly, CD276 was notably absent from non-tumoral vessels within the brain

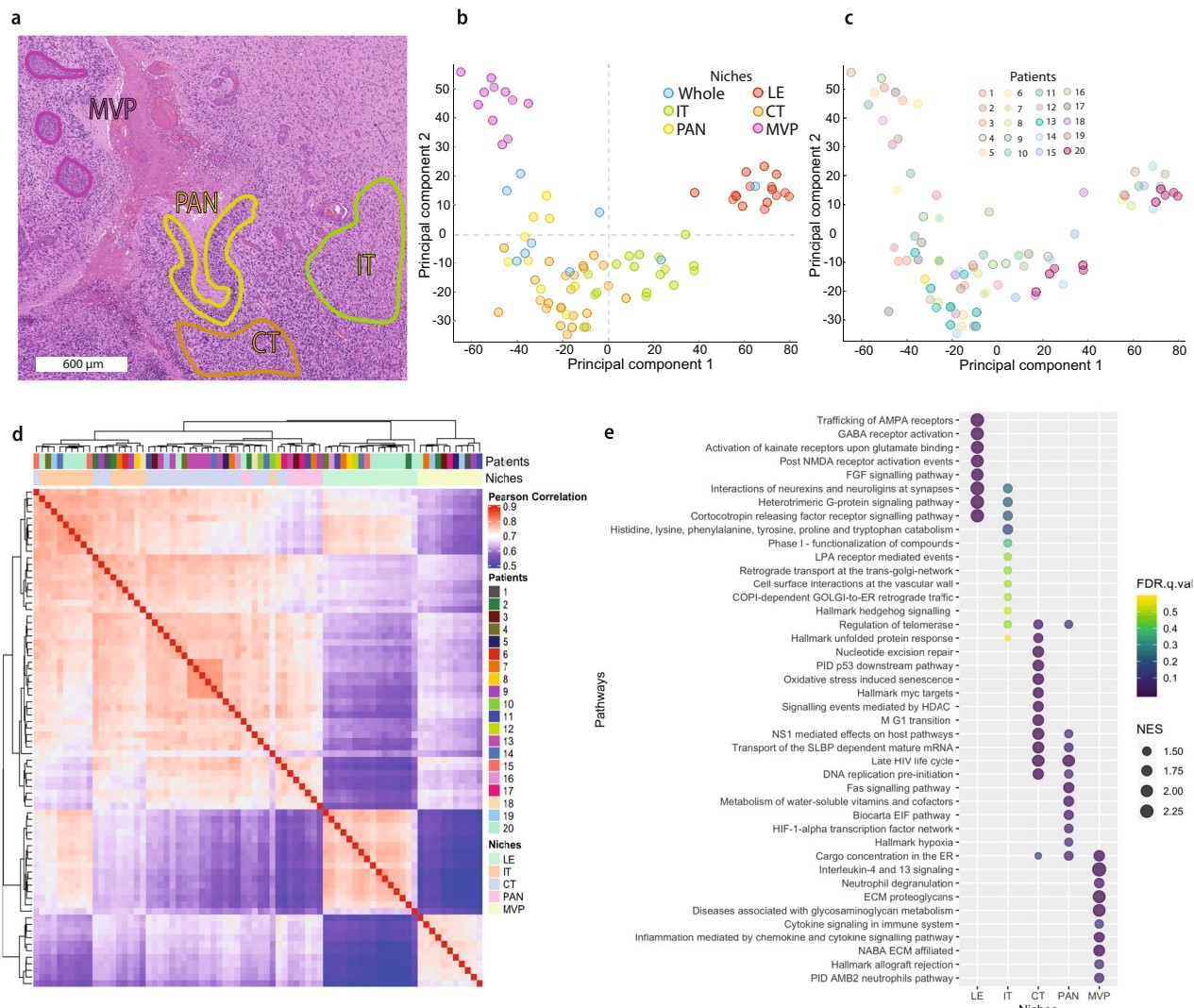

**Fig. 2 Gene expression in anatomic features. a** Annotated H&E section of GBM highlighting areas of palisading cells around necrosis (PAN), microvascular proliferations (MVP), infiltrating tumor (IT), and cellular tumor (CT) regions. **b**, **c** Multidimensional scaling of all proteins using principal component analysis highlight (**b**) region- and (**c**) patient-specific protein spatial distributions. Note the reduction of inter-patient variability of regional signatures compared to whole (bulk) tissue profiling. Overall, samples derived from the same anatomic feature, regardless of patient origin, were more similar to each other. **d** Pearson correlation within and between tumors based on hierarchical clustering of all proteins ($n = 78$ samples). Anatomical features are a strong driver of proteomic intra-tumoral heterogeneity in GBM. **e** Gene Set Enrichment Analysis (GSEA) is based on all samples and their comparisons against other sample types. Normalized enrichment score (NES) is derived from the GSEA output and accounts for differences in gene set size and in correlations between gene sets in the expression dataset. Source data are provided as a Source Data file.

parenchyma, suggesting a potential contribution to the immunosuppressive GBM microenvironment[34] (Fig. 3i). These results are supported by the observed unequal distribution of T infiltrating lymphocytes in some GBM and the role of CD276 in suppressing T helper cell immune response in other models[35,36]. While we found concordant staining patterns of CD276 and AKAP12 within the HPA[37], we believe such regional markers could be underappreciated in samples that lack adequate sampling of PAN or MVP regions highlight the usefulness of this approach and resource for defining relevant niche-specific markers. For instance, previous studies have demonstrated through regional single-cell mapping[38] and immunocytochemistry[39] the relative depletion of T cells within GBMs, especially in MVP regions and associated hypoxia[35], compared to other malignancies including less invasive gliomas.

In addition to proteogenomically concordant markers, we also orthogonally validated the presence of protein tyrosine phosphatase receptor type Z1 (PTPRZ), a candidate only found to be

enriched in IT regions at the protein level (Fig. 3j). Indeed, PTPRZ1 showed strong staining in infiltrating cells (Fig. 3k–l) and has been recently implicated in tumor invasion and aggressiveness[40–42].

**Regional intra-tumoral heterogeneity in GBM exists along a KRAS-MYC-hypoxia axis.** Our initial analysis focused on using the histologically defined hallmark regions of GBM to guide proteomic comparisons. It is, however, possible that additional layers of regional biological heterogeneity exist in GBM that are not solely defined by these anatomical features. To capture activation levels of distinct molecular programs at the individual sample level, independently of regional group memberships, we trained and applied machine learning models that could infer the activation level of functionally relevant gene sets in our dataset. To do this, the thousands of gene sets found in the Molecular Signatures Database (MSigDB)[43], which were developed in a myriad

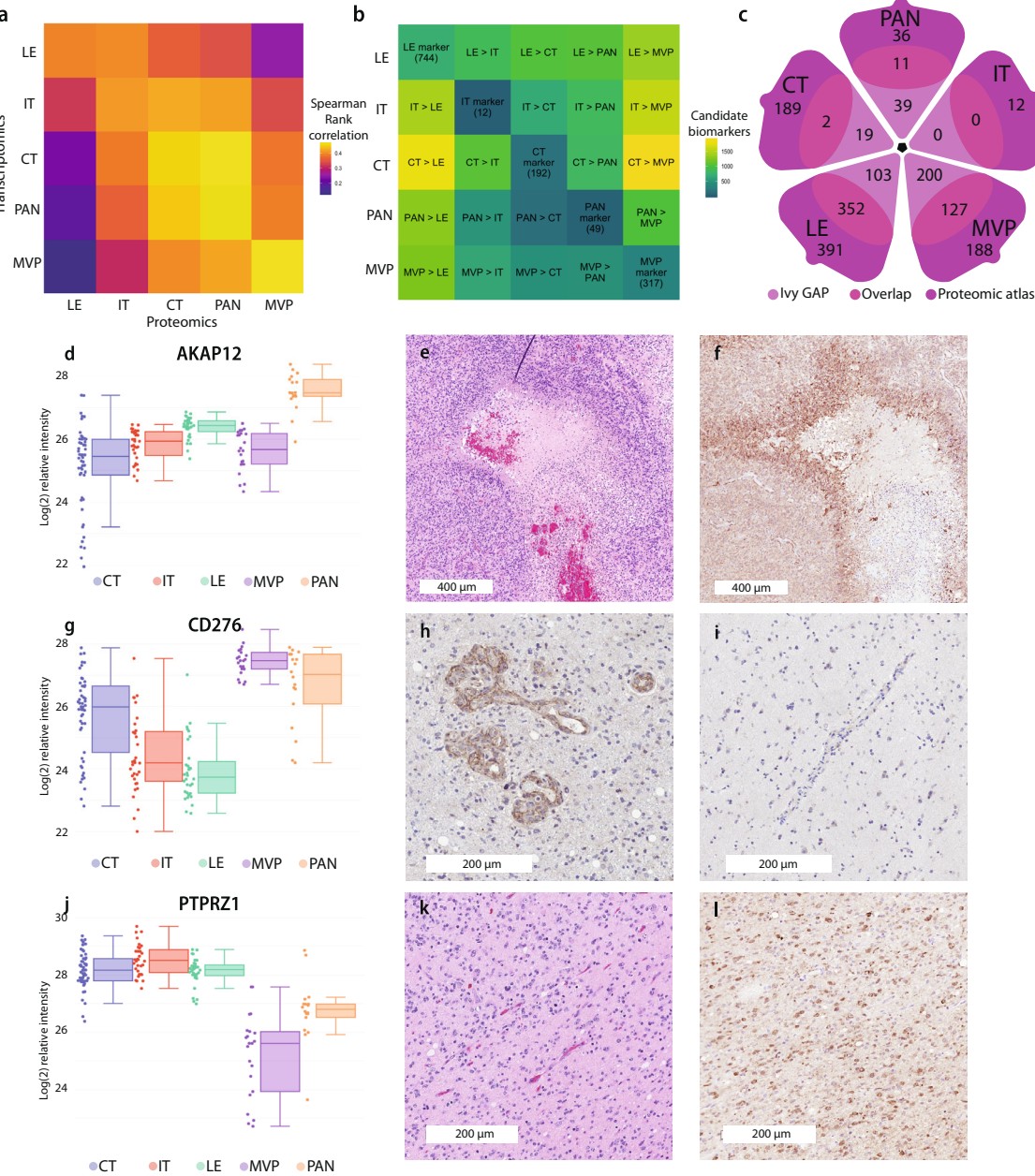

**Fig. 3 Region-specific biomarker identification and validation. a** Spearman Rank correlation between histomorphological transcriptomic (Ivy GAP) and our own proteomic profiles, leading-edge (LE), infiltrating tumor (IT), cellular tumor (CT), palisading cells around necrosis (PAN), and microvascular proliferations (MVP). Only genes that were identified in both datasets were included in the analysis. **b** Differential expression matrix analysis based on genes identified based on all samples based on a two-sided t-test at $p < 0.01$ and 76 degrees of freedom. Values along the diagonal are the number of genes that are differentially abundant against all other regions. **c** Differential expression matrix comparison by Venn diagram across transcriptionally (Ivy GAP) and proteomically significant biomarkers. Only genes that were identified in both datasets were included. **d** Regional enrichment of AKAP12 by boxplot highlights increased expression within PAN ($n = 154$). Data are presented as median values ± IQR and min/max values (whiskers). **e** H&E image of GBM highlighting necrosis, PAN, and CT. **f** Region-specific validation within an external cohort IHC demonstrates high staining of AKAP12 within PAN. **g** Regional enrichment of CD276 using boxplot highlights increased expression within MVP ($n = 154$). Data are presented as median values ± IQR and min/max values (whiskers). Region-specific validation within an external cohort by IHC demonstrates (**h**) high staining of CD276 within MVP but (**i**) no staining of vasculature in LE. **j** Regional enrichment of PTPRZ1 by boxplot highlights increased expression within IT ($n = 154$). Data are presented as median values ± IQR and min/max values (whiskers). **k** H&E image of GBM highlighting LE and IT. **l** Region-specific validation within an external cohort by IHC demonstrates high staining of PTPRZ1 within tumor cells of the IT. Source data are provided as a Source Data file.

of tissue types, pathologies, and model organisms, were distilled to a subset that is relevant to GBM niches. Firstly, ssGSEA scores were calculated for each signature at the protein level (this study) and RNA level (Ivy Gap study); secondly, we trained XGBoost regression models that take as input protein or RNA expression data, respectively, to infer the ssGSEA score of each signature; and thirdly, a set of 64 signatures was selected, each of which produced concordant predictions from matched protein and RNA

expression data from a third independent GBM cohort, as described in the Methods section (Supplementary Data File 4). We make this valuable set of XGBoost models to infer the level of activation of these 64 functional programs available for downloading (https://github.com/diamandis-lab/paper-prot-atlas-gbm) and via an online tool (https://cancerhub.shinyapps.io/prot-atlas-gbm/).

For this study, we used this 64 signature set to explore the functional landscape of GBM in a region-agnostic and unsupervised manner. For this analysis, we chose to focus on only CT and PAN, as these regions contained the highest proportion of pure tumor cells (nearly 100% by histology) compared to other regions like MVP, IT, and LE that contained a significant proportion of non-neoplastic cellular elements that could confound the analysis (Supplementary Figs. 2–25). Hierarchical clustering of these purified tumour tissue regions (CT and PAN samples) (Fig. 4a) revealed the presence of two functional groups independent of the samples' regional annotations, where these clusters lacked significant enrichment in CT and PAN samples, ($\chi^2 = 0.021$, $p = 0.886$), and were further validated by t-distributed Stochastic Neighbour Embedding (tSNE) analysis (Supplementary Fig. 28). Moreover, this clustered structure also appeared to be independent of tumor origin with samples belonging to different niches from the same patient often segregating to different clusters (Fig. 4a). To define the functional characteristics of each cluster, we analyzed the contribution of the 64 gene sets to the cluster structure using the R package "psSubpathway", revealing 26 signatures significantly enriched in the samples of cluster 1, including the GBM mesenchymal (Verhaak), cell migration and KRAS targets signatures (Fig. 4b and Supplementary Data File 4). Cluster 2 was characterized by a different set of 11 enriched signatures, including the GBM proneural (Verhaak), embryonic stem cell, and MYC targets signatures.

Closer inspection of signature enrichment patterns revealed an inverse correlation between the KRAS targets and MYC targets signatures (Fig. 4c), and, despite being intermixed between CT and PAN regions, these anatomical coordinates appeared to be relatively further faithfully segregated across a third hypoxia molecular signature axis that was also one of the concordant proteogenomic programs defined by our analysis (Fig. 4d). This triple-axis of separation was validated by interrogation of the independent IVY gap transcriptional atlas (Fig. 4e), and confirming, within the CT samples, that high KRAS targets activity is associated with invasion and epithelial-to-mesenchymal transition processes whereas samples enriched for the MYC axis were associated with cell cycle progression (Fig. 4f).

Of note, the Verhaak_GBM_classical signature was not included in our set of 64 proteogenomic signatures due to low protein/RNA concordance (robust linear fit's $p = 0.059$). Notably, both this signature and the neural subgroup (which did have good protein-RNA correlation) did not meaningfully correlate with the two defined protein-based clusters. Given the aforementioned proteogenomic discordances in cancer programs and contextual differences (e.g., bulk transcriptomics vs. regional proteomics), we used the area under the receiver operator curve (AUC, ROC) to define the nomenclature of our spatially-profiled and protein-based clusters (Fig. 4g). Importantly, in our proteomic datasets, MYC and KRAS gene set-driven grouping yielded more robust sample separation than Verhaak Mesenchymal and Proneural gene sets (Supplementary Fig. 29). In addition, our defined axes had more refined and restricted molecular modules offering a more precise and narrow biological correlate of the defined subgroups (Supplementary Fig. 30). We, therefore, use KRAS- and MYC-signature sets to further characterize our data.

Radar plots were used to visualize the relative contributions of the different molecular axes within individual patients. This revealed many examples in which regional profiles from the same tumor were highly divergent from one another and the bulk tumor (Supplementary Fig. 31). Together, this unsupervised analysis provides support for the existence of a triple-axis, defined by KRAS_targets, MYC_targets, and Hypoxia patterning regional aspects of heterogeneity in GBM. Importantly, the relative coordinates of a GBM sample along this KRAS-MYC-hypoxia model were associated with distinct functional profiles.

**Single-cell RNA (scRNA) profiles in GBM support the role of functional KRAS/MYC divergence as source of intra-tumour variability**. To further characterize intra-tumour variability, the enrichment level of the KRAS_TARGETS and MYC_TARGETS signatures was assessed in a group of samples at the single-cell level by using RNAseq data from recent studies[44]. GBM profiles formed three groups defined either as KRAS_TARGETS-high, MYC_TARGETS-high, or belonging to a Central group that shows no elevated enrichment in either signature (Fig. 4h and Supplementary Fig. 32). Importantly, cells highly enriched for both signatures are nearly non-existent, supporting the observed mutually exclusive feature of these cellular phenotypes. Similar to the regionally micro-dissected samples, the MYC_TARGETS-high group presented high enrichment of the HALLMARK_MITOTIC_SPINDLE signature (Fig. 4i) whereas KRAS_TARGETS-high cells were enriched in WU_CELL_MIGRATION, a signature that is associated with tumour invasiveness (Fig. 4j). These results support a model where a large proportion of GBM cells within a tumour tissue appear to be oncogenetically stable, with certain subpopulations exhibiting "oncogenic activation" towards either mutually exclusive invasive or proliferative phenotypes. At the individual sample level, we noticed cells can exhibit a strong dominance towards one of these phenotypes, or contain a mixture of both invasive and replicating GBM cells (Fig. 4h and Supplementary Fig. 31). This observation is also supported within our cohort and could represent inter-tumoral heterogeneity pressures and/or be explained by sampling approaches in different studies.

**A machine learning strategy for niche-specific gene signature inference**. Variability in gene expression due to varying levels of tumour purity along with different proportions of histomorphologic niches is a widely acknowledged limitation, albeit rarely addressed, in many proteomics and transcriptomics studies. Our data demonstrates how the presence of non-tumour cell tissue can greatly influence the inferred activation levels of gene signatures (Supplementary Fig. 33). For instance, the level of enrichment in the KRAS targets signature presents extreme levels in the MVP niche, and is probably responsible for raising global levels of this signature in the whole tissues (Supplementary Fig. 34a); on the other hand, enrichment in the MYC targets signature is lowest in the non-tumour compartments LE and MVP, probably dampening the observed signal in bulk tissue taken near the infiltrating edge (Supplementary Fig. 34b). The hypoxia signature shows highest levels in the PAN niche, as expected (Supplementary Fig. 34c); nonetheless, the fact that the observed levels in the whole tissue are not grossly deviated from those in the tumour niches does not rule out bias introduced by the non-tumour fractions. These results highlight the importance of accounting for tumour composition when working with whole tissue samples.

We sought to develop a computational strategy to infer CT-niche-specific signatures that produces equivalent results to ssGSEA scores when used in histologically pure samples, but that is less prone to interference from other niches present in bulk GBM samples (Fig. 5a). Data from histomorphologically defined

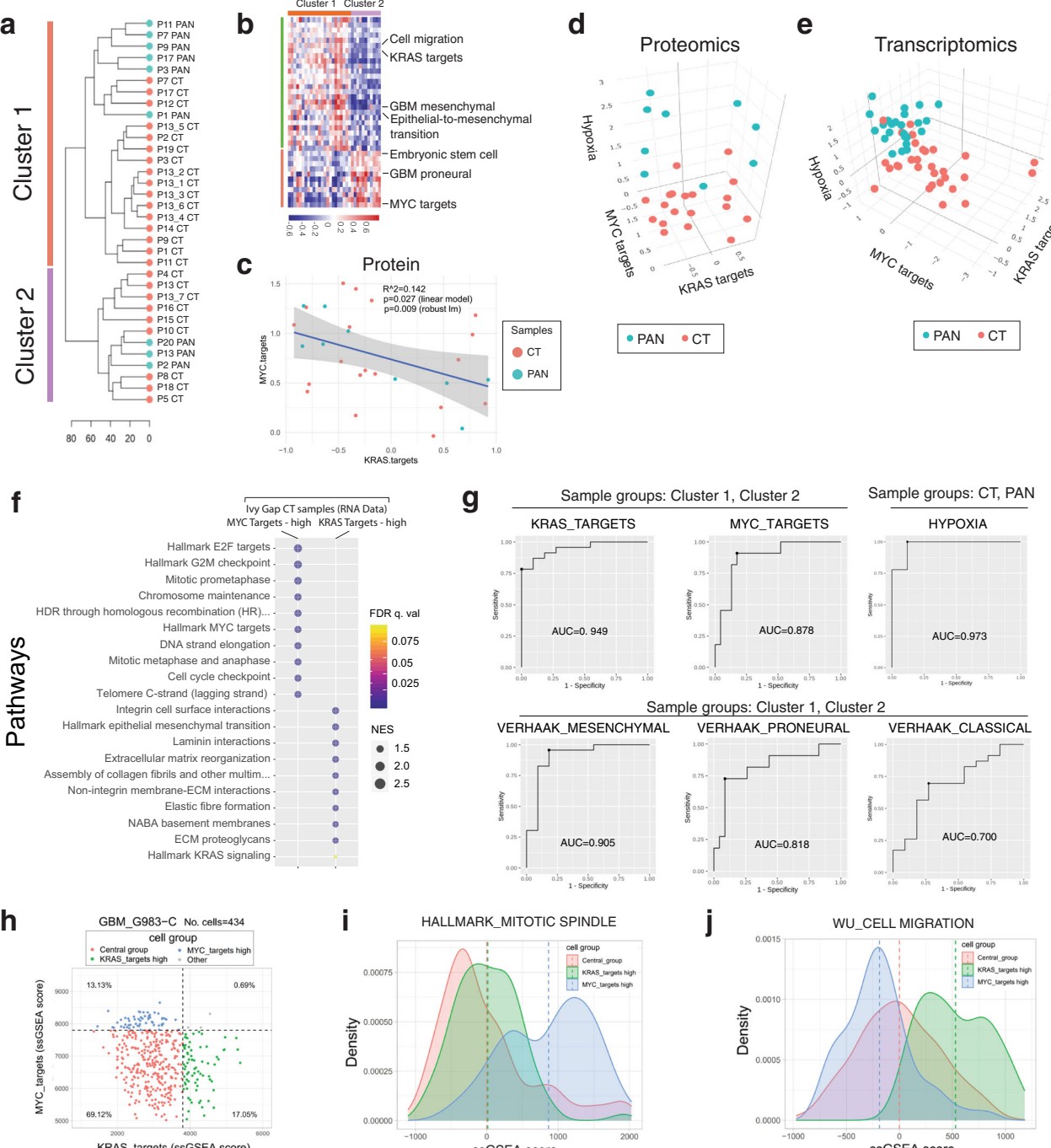

**Fig. 4 Single-sample gene signature analysis of highly pure GBM tumour fractions reveal two biologically distinct clusters. a** Two clusters of samples, independent of histomorphologic niches, are revealed by hierarchical clustering of samples using the ssGSEA scores from 64 selected gene signatures. Teal represents palisading cells around necrosis (PAN) samples while orange represents cellular tumour (CT) samples. **b** Functional drivers associated with each cluster using psSubPathways software package. **c** Inverse correlation between the KRAS targets and MYC targets signatures in the protein dataset generated in this study ($n = 34$ samples); additional null hypothesis testing was performed by permuted linear model fit analysis (lmPerm R package, $p = 0.024$). The 95% confidence interval is shown as grey areas. **d** 3D scatter plot of ssGSEA values of KRAS targets, MYC targets, and hypoxia gene signatures using proteomics (this study). **e** Transcriptomics (Ivy Gap) data from histomorphologically-defined samples. **f** Functional profiling by GSEA of Ivy Gap CT samples for the groups of samples labeled as MYC targets-high ($n = 9$) vs KRAS targets-high ($n = 5$) while presenting low activation of the hypoxia signature. **g** ROC analysis of clustering based on KRAS_targets, MYC_targets, or HYPOXIA gene signatures compared to that using Verhaak et al. mesenchymal, proneural and classical gene sets using cut-off optimization with the Youden method. **h** Distribution of KRAS_targets and MYC_targets signature enrichments in GBM at the single-cell level (Richards et al.) where (**i**) Mitotic spindle enrichment is quantified across the three detected populations. **j** Cell migration process enrichment across the three detected populations. Source data are provided as a Source Data file.

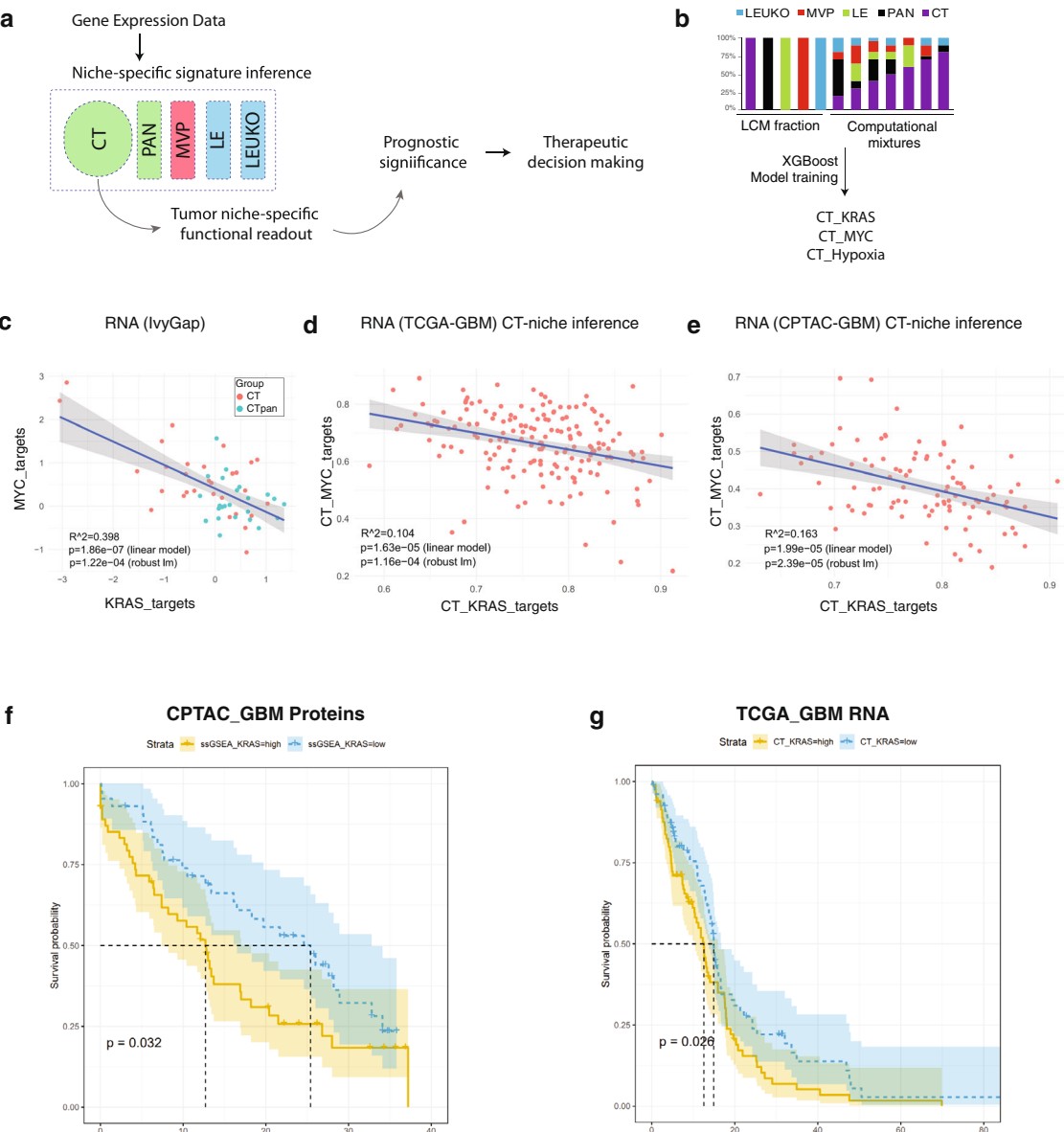

**Fig. 5 High levels of the "KRAS targets" signature is associated with clinical aggressiveness in GBM patients. a** Schematic depicting non-tumour tissue content influencing overall molecular signatures and how computational methods that allow for niche-specific inference may lead to robust actionable insights. **b** A synthetic dataset was built from pathologically defined leading edge (LE), cellular tumor (CT), palisading cells around necrosis (PAN), and microvascular proliferations (MVP) tissue samples by generating computationally simulated mixtures; this dataset was used to train XGBoost models to infer the KRAS targets, MYC targets and hypoxia signatures in a CT-niche specific manner. **c** Association between KRAS and MYC target signatures in the RNA Ivy Gap dataset ($n = 54$). Null hypothesis testing was performed by permuted linear model fit analysis. The 95% confidence interval is shown as grey areas. **d** CT-niche-specific inference in the TCGA-GBM dataset ($n = 171$). Null hypothesis testing was performed by permuted linear model fit analysis. The 95% confidence interval is shown as grey areas. **e** CT-niche-specific inference in the CPTAC-GBM dataset ($n = 100$). Null hypothesis testing was performed by permuted linear model fit analysis. The 95% confidence interval is shown as grey areas. **f** Kaplan-Meier survival curves for samples split into "high" and "low" KRAS targets activity using ssGSEA of CT-niche specific inference in a proteomics dataset (CPTAC) and (**g**) at the RNA level (TCGA-GBM). Source data are provided as a Source Data file.

samples was used to generate synthetic computational mixtures with known niche proportions together with their associated signature enrichment scores and subsequently used to train XGBoost regression models for RNA expression data (Fig. 5b, Supplementary Fig. 35), as detailed in the methods section. The inverse correlation between the signatures KRAS targets and MYC targets that are observed in the RNA samples of the Ivy Gap study by ssGSEA (Fig. 5c) was also confirmed by CT-specific inference in the RNA data from bulk samples of the TCGA-GBM (Fig. 5d) and the CPTAC-GBM studies (Fig. 5e, Supplementary

Fig. 36). Of note, we did expect to get a more subtle inverse relationship between the two axes in the TCGA (and CPTAC) datasets, as compared to IvyGAP, given that the data is derived from bulk samples. Even when inferring the make-up of the CT compartment, it is likely that CT-inferred patterns of the TCGA and CPTAC samples are derived from multiple and diverse CT-niches (combination of MYC and KRAS niches) dampening the more pronounced region-to-region differences we observed in both our dataset and that of the IvyGAP. Finally, by using this inference model, we show that high levels of CT-specific KRAS

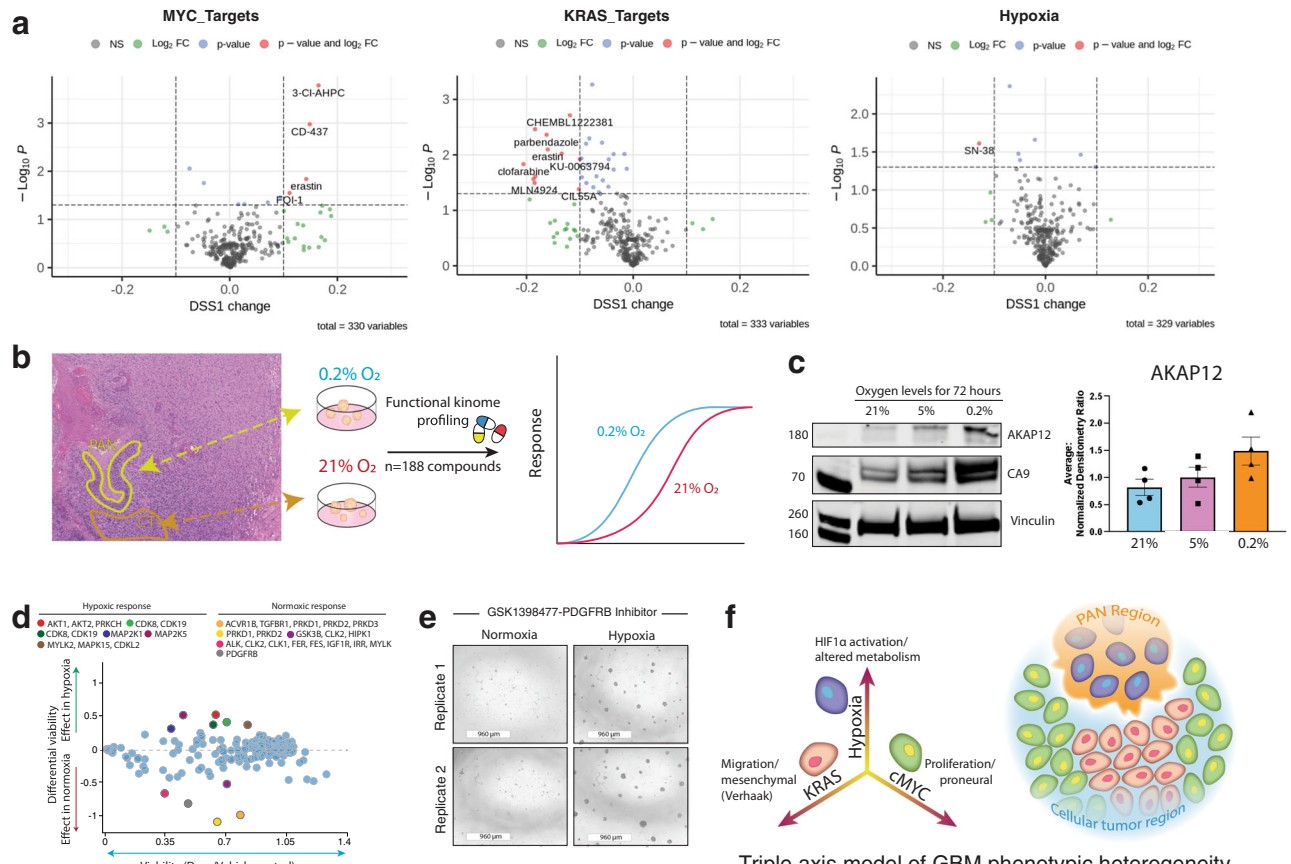

**Fig. 6 Pharmacological profiling reveal axis-specific drug vulnerabilities and resistance. a** Volcano plots showing pharmacotranscriptomic comparisons of drug sensitivities across cells lines ranking high and low along with the MYC-, KRAS- and hypoxia axis. **b** Schematic overview of pharmacological profiling of hypoxic and normoxic cell populations to highlight potential differences in drug sensitivities along this axis (left). **c** Western blots analysis highlights increased CA9 expression with decreasing oxygen levels indicating a downstream response to hypoxia. Relative expression of palisading cells around necrosis (PAN) associated marker AKAP12 by western blot reveals increased expression in hypoxic conditions. Vinculin serves as a loading control. Densitometry analysis of AKAP12 western blots highlights an average 2-fold increase between 21% oxygen and 0.2% oxygen conditions across 4 different cell lines. **d** Differential viability of hypoxic versus normoxic cell populations (y-axis) upon treatment with 188 compounds and averaged over two replicates, overall cell viability is the average of the cell populations against the reference population (x-axis) and targets with the greatest differential viability are colored by the target as specified in the top legends. Blue dots represent compounds with the minimal differential response. **d** Relative viability effects on GSC proliferation in a kinome screen under differential oxygen concentrations. **e** Spheroid images upon treatment of GSK1398477, a PDGFRB inhibitor, in hypoxic and normoxic conditions. **f** Model of the heterogeneous co-existence of GBM populations driven by the KRAS or MYC target genes and signatures defined on the hypoxia pathway axis. Source data are provided as a Source Data file.

targets signature were associated with a more aggressive clinical phenotype with a shorter overall survival period in the CPTAC cohort (Fig. 5f). This association was also confirmed at the RNA level in the TCGA-GBM cohort by ssGSEA (Fig. 5g).

**Pharmacotranscriptomic and chemical profiling highlight drug sensitivity differences along the three axes of GBM heterogeneity.** Finally, we explored the therapeutic significance of the triple-axis model to understand if the observed regional differences could potentially drive treatment resistance and affect patient-level therapeutic decisions. Specifically, we compared drug sensitivities differences of 543 drugs in the CTRPv2 dataset across the upper and lower quartile of GBM cells grown in culture ($n = 31$) ranked by their relative enrichment for axis-specific signatures (Supplementary Data File 5). By using these GBM cell lines as experimental avatars of regional phenotypes, we show axis-specific sensitives and resistance to numerous agents (Fig. 6a). Specifically, we found a number of traditional anti-cancer agents showed biological activity only in MYC-enriched cell lines (e.g., 3-CI-AHPC, CD-437). This is perhaps expected

given the enrichment of pro-proliferative programs found in MYC-enriched tumor areas. Conversely, ranking cell lines based on their KRAS-axis activation showed the opposite result with relative resistance to probes targeting many classic cancer targets (e.g., Parbendazole). This heterogeneity of drug sensitivities along this MYC-KRAS axis may help explain the observed clinical aggressive phenotype of KRAS-enriched tumors and provide insight into treatment resistance. Furthermore, it suggests that cancer cells along the MYC-RAS axis may require distinct profiling approaches and cell line avatars to discover axis-specific therapies. Importantly, despite the correlation of our protein-based KRAS and MYC axes with the Veerhak mesenchymal and proneural transcriptional subtype programs, we note that drug sensitivities were severely blunted and absent from the latter classification system (Supplementary Fig. 37). These differences, with potentially important clinical implications to drug design and selection, were felt to be attributed to relatively refined and focused gene modules that define the KRAS and MYC axes (Supplementary Fig. 32).

Ranking cell lines based on their relative hypoxia signature only revealed SN-38, a topoisomerase inhibitor, as an agent that

showed differential effectiveness along this axis. We reasoned the relative lack of differential sensitivity could be explained by the uniformly high and non-physiological oxygen concentration that these cell lines were grown and profiled under. This limitation may therefore under-represent the relative contribution of hypoxia to pharmacological resistance in this analysis. To test if we could model additional drug sensitivity differences for cells along the hypoxia axis, we grew and chemically profiled patient-derived glioma stem cell-like (GSCs) cells in high and low oxygen concentrations (21% and 0.2% $O_2$) (Fig. 6b). Western blot analysis confirmed a time-dependent activation of the hypoxia marker carbonic anhydrase IX (CAIX) and AKAP12 after 72 h of culture, suggesting that these culture conditions could recapitulate, at least, some aspects of the hypoxic niche we defined in our clinical tissue samples (Fig. 6c). Following an initial climatization of the GSCs in the different oxygen concentrations, we concurrently profiled their relative sensitivity to a library composed of 188 kinase inhibitors with Alamar blue as a readout of cell growth (Fig. 6d and Supplementary Data File 6)[45]. Similar to the MYC- and KRAS- axes in our computational analysis, some drugs did indeed show distinct responses based on oxygenation levels (Fig. 6d). For instance, PDGFRB inhibitor significantly impaired GSC viability only in normoxic conditions, with little effect in hypoxia (Fig. 6e). Conversely, other agents show higher potency under hypoxic conditions (Supplementary Fig. 38). Together, these findings support that in addition to the MYC-enriched and KRAS-enriched programs, microenvironmental pressures can also drive dynamic changes in regional biology of GBM that may blunt or alter their sensitivity to therapeutic agents (Fig. 6f).

## Discussion

Molecular profiling continues to refine our biological understanding of GBM to include multiple layers of inter- and intra-tumor heterogeneity. Regional transcriptional profiling by the Ivy GAP initiative has recently facilitated the compartmentalization of molecular information to the classic hallmark cellular niches that define this deadly disease. Here, we further these efforts by overlaying downstream proteomic phenotypes to GBM's cellular anatomy and highlight important implications for precision and personalized therapy design.

Our study notably offers a number of distinguishing features and complementary insight from existing resources and initiatives. Compared to previous bulk and regional gene-based profiling studies, we focus on protein-level information. While these molecular species are often closely related, there is now substantial evidence that their correlation is modest and can lead to the nomination of different molecular pathways and therapeutic targets when considered in isolation[28]. Here, we show how molecular targets can be prioritized based on regional co-enrichment at both RNA and protein levels (eg. AKAP12, CD276), or discordantly enriched only at the protein level (PTPRZ1). These consistently elevated molecular components may reflect generalizable and reliable targets for niche-specific therapies. Alternatively, an exploration into proteogenomic discordant patterns could provide important insight into the post-translational regulatory mechanisms in GBM. Our resource also provides complementary value to other existing proteomic datasets. Firstly, compared to IHC approaches, our strategy offers quantitative information allowing for discrimination of more subtle regionally enriched proteins. Similarly, we complement recent deeper phosphoproteomic initiatives of frozen tissue specimens by allowing deconvolution of pathways that may be operational in only focal areas of the tumor[11]. Our triple-axis model highlights how important patterns of potential

actionability could have been otherwise overlooked by either bulk profiling or tissue microarray approaches.

To further highlight the utility of our resource, we validate the selective localization of the immune checkpoint protein CD276 (B7-H3) to tumor vasculature (MVP), information that may help explain and overcome challenges with immunotherapies focusing only on canonical proteins such as PD-1 and CTLA4[46]. CD276 has been identified to be highly expressed within the vasculature of other cancers and has been targeted using antibody-drug conjugates[37], chimeric antigen receptor (CAR) T cell therapy[33] and monoclonal antibodies (MAB)[47]. Its confirmed expression within GBM suggests that similar anti-CD276 approaches may also benefit future GBM immunotherapy trials. Similarly, GBM shows geographic areas of hypoxia that continue to be implicated in promoting cancer stem-cell-like behavior and resistance to conventional chemoradiation therapy[48]. Our resource was able to highlight a number of PAN-enriched proteins, including AKAP12. While relatively understudied in GBM, AKAP12 has been shown to be induced by hypoxia in melanoma where it contributes to tumor growth and spread[32]. Our data support that AKAP12 may play a similar role in GBM.

In addition to its utility as a unique resource, integrating our findings with existing molecular profiling datasets allowed us to propose and validate a model where GBM intra-tumor heterogeneity can be defined to exist across a three-dimensional axis defined by their relative MYC-, KRAS- and hypoxia programs. Of clinical importance, we show, across a number of datasets, tumors showing high activity and relative proportions for the KRAS-axis, using both transcriptomic and proteomic readouts, were clinically more aggressive. Routine quantification of this axis could therefore provide more precise prognostic data for patients with GBM. Furthermore, this triple-axis model provides insight into how we could understand and approach treatment resistance and failure. For instance, by aligning these three proteogenomic programs to drug sensitivity profiles in large pharmacogenomic datasets, we show that this molecular heterogeneity of GBM can affect cellular responses. This includes relative sensitivities and resistance to various existing chemical agents across the different axes that could require combination cocktails to adequately address each compartment. The addition of some conventional anti-cancer agents to cell lines with prominent MYC-programs results in reduced cell growth. Conversely, we find that many compounds are relatively ineffective in targeting cells enriched for KRAS_-targets molecular signature. This relative blunted the effect of conventional agents for this program could explain the more clinically aggressive nature of KRAS-enriched tumors. We further substantiate the regional effects of the micro-environment to drug response by carrying out a pan-kinome chemical screen in normal and low oxygen culture conditions. This experiment highlighted that dynamic movement of genetically similar cells along a specific axis could also alter drug responses to targets found in our existing models of GBM (e.g., PDGFR). Encouragingly, there is also relevant correlations of our proposed axes with previously described transcriptional profiles found to be variable at both the inter-patient and intra-patient level. It is important to note that observed differences represent drug sensitivities under conventional cell culture conditions and may be different from those observed in more physiological hypoxic conditions. Using this as a framework, future studies recapitulating molecular signatures of MYC- and KRAS-targets within cell line avatars could further explore the simultaneous contributions of these axes on pharmacological targeting. Notwithstanding these limitations, we believe our data could have important clinical utility by providing more refined protein-level molecular axes that define GBM heterogeneity and could guide future drug development efforts. These findings emphasize the importance of considering regional

protein-level dependencies and vulnerabilities to simultaneously target phenotypically distinct tumor sub-compartments and overcome resistance.

Our approach has a number of potential limitations that should be considered when utilizing this resource. While the use of FFPE tissue allowed us to reliably define GBM hallmark features for micro-dissection with high fidelity, cross-links formed during formalin fixation, along with the relatively small size of certain niches (PAN and MVP), limited deeper proteomic analysis using sample fractionation. In addition, it is possible that our relatively smaller cohort size of 20 patients may not recapitulate the full repertoire of proteomic patterns that exist across patient groups in this highly heterogeneous tumour type. Notwithstanding these limitations, we highlight complementarity and non-overlapping value with the other GBM resources available in the community. By leveraging bioinformatic approaches and hallmark pathways/processes, defined by modules of gene/proteins sets, we could reliably overcome this limitation to define relevant regional pathway-level differences and drug candidates even within individual patient samples. Despite undeniable strengths of ssGSEA in estimating the activation status of biological programs, the use of machine learning is a very active area of research leading to development of OMICs-based applications, such as inferences of drug susceptibility based on transcriptomic profiles. In this study, XGBoost served to learn molecular patterns of interest in a well-characterized dataset that could be later extrapolated to infer the functional status of samples from other datasets. The values inferred by the XGBoost models bear the context of the training sets, as these come scaled to a reference framework and less prone to interferences from non-target biological processes. Here, we, therefore, use ssGSEA as a tool to establish the ground truth across relevant datasets and apply machine learning tools to aid in extrapolation of the most relevant information to other datasets. While our niche-specific infererence model was initially developed on LFQ proteomic data, a recent study highlighted how TMT datasets could be used to generate LFQ-equivalent values[49]. This inference model, in theory, could therefore also be extended to other TMT-based datasets in future studies. This exciting prospect is especially supported by the comparison of ssGSEA proteogenomic concordances of relevant programs we highlighted in this study (e.g., KRAS/MYC/Hypoxia, Supplementary Fig. 36).

Together, this study topographically overlays proteomic patterns onto the hallmark histomorphologic features of GBM. We host the entire dataset in an interactive portal allowing users to explore and query the resource in real-time (https://www.brainproteinatlas.org/dash/apps/GPA). Furthermore, we use the generated dataset to define and validate a subset of promising regional markers for future studies, and a unifying triple-axis model that captures important and clinically relevant aspects of inter- and intra-tumoral GBM heterogeneity. By resolving the relevant spatial phenotype-level biology and how it varies within patient's tumors, we hope this work will accelerate our biological understanding and therapeutic efforts for this heterogenous and deadly disease.

## Methods

**Ethical approval**. The University Health Network Research Ethics Board has approved the study REB #17-6193 as it has been found to comply with relevant research ethics guidelines, as well as the Ontario Personal Health Information Protection Act (PHIPA), 2004. Patient consent was not directly obtained and a consent waiver for this study was granted by the University Health Network Research Ethics Board as the research was deemed to involve no more than minimal risk as it included the use of exclusively existing pathology specimens.

**Study design**. We leverage mass spectrometry to develop a human glioblastoma atlas that aligns proteomic patterns to hallmark histomorphological features and

highlight niche-specific phenotype-level biomarkers and targets. The cohort consisted of 25 patients (Supplementary Data File 1), 20 for discovery and 5 additional cases for immunohistochemical validation. For discovery, tumors from the 20 patients were used to generate 990 H&E sections mounted on PEN slides. For each case, the selection of anatomical niches was standardized by independent annotations provided by two board-certified neuropathologists (M.R., P.D.). TME differences across patient samples were normalized using the same selection criteria as IvyGAP. Leading Edge (LE) is the outermost boundary of the tumor, where the ratio of tumor to normal cells is about 1–3/100, and the laminar architecture of the cortical layers is frequently evident. Infiltrating Tumor (IT) is the intermediate zone between the Leading Edge (LE) and Cellular Tumor (CT), and is frequently marked by perineuronal satellitosis. Cellular Tumor (CT) constitutes the major part of the tumor core, where the ratio of tumor cells to normal cells is between 100/1 to 500/1. Pseudopalisading Cells around Necrosis (PAN) are the narrow boundary of cells arranged like pseudopalisades along the perimeter of necrosis. Microvascular Proliferation (MVP) refers to two or more blood vessels sharing a common vessel wall of endothelial and smooth muscle cells arranged in the shape of a glomerulus or garland of multiple interconnected blood vessels. The LCM area cutoff for every anatomical niche and for each patient to be included for proteomic analysis was 40,000,000 $\mu m^2$ to ensure sufficient proteomic coverage. Mass spectrometry-based proteomics was then carried out for each sample within this cohort. A technical replicate was performed for each sample, such that the sample was non-sequentially injected twice into the mass spectrometer. Outlier analysis (cook's distance and leverage plots) was performed utilizing the MaxQuant output on the number of proteins identified by MS/MS and the total number of proteins identified (Supplementary Fig. 1). Samples that were identified as outliers were removed from downstream analysis.

**Tissue processing for laser capture microdissection**. Formalin-fixed paraffin-embedded (FFPE) tissue blocks were sectioned at (10 μm thick) and mounted onto Leica PEN slides (Cat No. 11505189). Slides were deparaffinized using 100% xylene (2x), 100% ethanol, 95% ethanol, 70% ethanol and 50% ethanol (3 min each). This preparation was followed by hematoxylin staining (Vector) for 1 min. Slides were then rinsed in de-ionized water (1 min) and stained in 1% eosin Y (Fisher scientific) and 1% calcium chloride (6 min). Slides were then left to air dry for laser capture microdissection.

**Laser capture microdissection**. PEN slides were kept at room temperature for laser capture microdissection (LCM). Slides were treated with an anti-static gun (Sigma) prior to LCM. Stained sections mounted on PEN slides were micro-dissected for specific anatomical niches while referring to the annotated H&E images. A Leica LMD 70000 (Leica Microsystems, Inc., Bannockburn, IL) was used for LCM. Samples were then collected in an Eppendorf tube and stored at room temperature for further sample preparation.

**Sample preparation**. For proteomic analysis, 50 μL of 1% Rapigest™ (Waters Corp.) was added to each sample and stored overnight at 4 °C. 200 μL of a solution including dithiothreitol (8 mM), ammonium bicarbonate (50 mM), and tris-HCl (200 mM) was added to each sample. The samples were then sonicated on high with 30 s intervals using a Bioruptor Plus (Diagenode) for 30 mins on high. Solutions were then centrifuged at 12,000 × g for 10 mins and the supernatant was collected. The supernatant was then heated to 95 °C for 45 mins followed by 80 °C for 90 mins using a ThermoMixer (Eppendorf). For alkylation, 20 μL of iodoacetamide (300 mM) was added to each solution in the absence of light. 1 μg of trypsin / Lys-C mix (Promega) was added to each sample and left to react overnight at 37 °C and acidified with trifluoroacetic acid (TFA) at a final concentration of 1% prior to stagetip cleanup step.

**Mass spectrometry analysis**. Samples were desalted with Omix C18 (Agilent Technologies) tips based on the manufacturer's protocol. Peptides were eluted with 3 μL (0.1% formic acid, 65% acetonitrile) and diluted with 57 μL (0.1% formic acid in MS water). 18 μL of solution (2.5 μg of peptides) was loaded from an auto-sampler, EASY-nLC1200 system (Thermo Fisher Scientific, San Jose, California) running Buffer A (0.1% formic acid). The analytical column consisted of an EASY-Spray column ES803A (Thermo Fisher Scientific, San Jose, California) heated to 50 °C. Peptides were eluted from the column at 300nL/min with an increasing concentration of Buffer B (0.1% formic acid in acetonitrile) over a 60 min gradient. The liquid chromatography setup was coupled to a Q Exactive HF-X (Thermo Fisher Scientific, San Jose, California) with a spray voltage of 2 kV with a 60 min data-dependent acquisition (DDA) method. The full MS1 scan was from 400 to 1500 m/z at a resolution of 70,000 in profile mode with selection of top 28 ions for further fragmentation using the HCD cell. Fragment ions were detected in the Orbitrap using centroid mode at a resolution of 17,500. These were the MS parameters: MS1 Automatic Gain Control (AGC) target was set to $3 \times 10^6$ with maximum injection time (IT) of 100 ms, MS2 AGC was set to $5 \times 10^4$ with maximum IT of 50 ms, isolation window was 1.6 Da, underfill ratio 2%, intensity threshold $2 \times 10^4$, normalized collision energy (NCE) was 27, charge exclusion was set to fragment only $2^+$, $3^+$ and $4^+$ charge state ions, peptide match set to preferred and dynamic exclusion set to 42 (for 90 min method).

**Immunohistochemistry.** Formalin-fixed paraffin-embedded (FFPE) tissue blocks were sectioned at (4 μm thick) and mounted onto positively charged glass slides. Sections were deparaffinized using 100% xylene (2x), 100% ethanol, 95% ethanol, 70% ethanol, 50% ethanol. and distilled water (3 min each). Slides were then submerged into a sodium citrate buffer (dihydrate tri-sodium citrate (2.94 g), Tween 20 (0.5 mL) adjusted to pH 6.0 with HCl) and heated in a pressure cooker for 17 min. Slides were cooled for 10 min and then washed with PBS (2x). The slides were placed in a humidified chamber and BLOXALL (vector) was added for 10 min. Slides were washed with PBS (2x) and incubated with SuperBlock (Thermo Fisher Scientific) for 10 min. Slides were then incubated overnight at 4 °C in a humidified chamber with the specific antibodies (Supplementary Data file 7). Slides were then washed with PBS (3x). Slides were then incubated with secondary antibodies for 30 mins. Slides were then washed with PBS (3x). Staining was then performed utilizing the DAB/Metal concentrate solution (Thermo Fisher Scientific). Slides were then washed with PBS (3x), followed by hematoxylin staining (Vector) for 1 min. Solutions were dehydrated using sequential solutions of distilled water, 50% ethanol, 70% ethanol, 95% ethanol, 100% ethanol, and 100% xylene (2x) for 3 min each. A coverslip was then mounted using permount (Fisher Scientific).

**Statistical analysis.** Mass spectrometry raw data files were processed using MaxQuant Andromeda (version 1.5.5.1) search engine (www.coxdocs.org) against the Human Swissprot protein database (July, 2019 version). The mass spectrometry proteomics data have been deposited to the ProteomeXchange Consortium via the PRIDE[50] partner repository with the dataset identifier PXD019381 (Analysis of proteomic data was performed using a variety of biostatistical platforms Perseus) (www.coxdocs.org), R (www.r-project.org), Orange (https://orange.biolab.si/), GSEA (https://www.gsea-msigdb.org/gsea/index.jsp) and Cytoscape (https://cytoscape.org/). Based on the number of proteins identified by MS/MS and the total number of proteins identified, Cook's distance and leverage analysis were then performed using R scripts to identify outliers (Supplementary Fig. 1). To average the technical replicates for PCA and heatmap analysis (Fig. 2b–d), values were averaged except when one value was missing from a technical replicate in which the non-zero numerical value was taken. Proteins were filtered such that only those that appeared in at least 60% within a group were included. The raw values were Log2 transformed and non-valid values were imputed (downshift = 0.3, width = 1.8). Gene set enrichment analysis (GSEA) was used to define pathways enriched in each anatomical niche. These pathways were then used in custom R scripts to filter and visualize unique and common pathways associated with each anatomical niche. Differential expression matrix analysis by t-test utilized all samples and was performed using a custom R script to generate lists of genes that are differentially expressed against other regions when average values are greater than the comparison group and statistically significant ($p < 0.01$). The genes identified as markers were common across all comparisons. Boxplots were generated using (https://www.brainproteinatlas.org/dash/apps/GPA) and points include all samples. Permuted linear model fit analysis was performed using the lmp function of the lmPerm R package with the "Exact" method and 5000 permutations. Robust linear model fit analysis was carried out using the lm_robust function of the estimate R package.

**Western blot.** Cell lysates were washed twice with PBS then lysed in 100ul of RIPA buffer containing complete protease inhibitor cocktail (Pierce) on ice. Samples were then centrifuged at 14000 × g for 20 min at 4 °C. Then the supernatant was collected and was measured for its total protein concentration by Bicinchoninic (BCA) Protein Assay (Thermo fisher). Samples were then prepared with XT Tricine Running Buffer (Biorad) and dithiothreitol (DTT) and were boiled at 95 °C for 10 min. Proteins were resolved on NuPAGE 3*Nature Communications* thanks Moussab Harb, Hiroyuki Matsuzaki, Ki Tae Nam and Rosendo Valero for their contribution to the peer review of this work. 8% Tris-Acetate Protein Gels (Invitrogen) and were transferred to polyvinylidene difluoride membrane (Immobilon-P, Millipore) using the Mini Trans-Blot Electrophoretic® Transfer Cell (Biorad). Blots were blocked in Licor blocking buffer (LI-COR) for AKAP12 and CA9 blotting. Primary antibodies used were: AKAP12 (1:500, HPA006344, Sigma), CA9 (1:1000, Donated by Sylvia Pastorekova) and Vinculin (1:30 000, ab129002, Abcam). Secondary antibodies, IRDye 800CW and IRDye 680RD from LI-COR (1:10 000) were used to visualize the protein of interest using a LI-COR Odyssey Image Studio (LI-COR).

**Selection informative gene expression signatures with RNA/protein concordance in GBM.** The ssGSEA algorithm allows inference of the activation status of gene signatures at the sample level, but it requires careful selection of the signatures to ensure that they are biologically meaningful in the field of study and that they present a low level of interference from overlapping signatures. Instead of using a "signature pruning" approach, we screened the MSigDB database to identify signatures that are suited for ssGSEA analysis in GBM samples.

From the proteomics dataset generated in this study, a total of 76 samples from the histopathologically defined niches CT, IT, LE, PAN, and MVP were selected, whereas the group of WHOLE samples was excluded to avoid possible signature contamination due to the presence of multiple niches in the same sample. To

reduce the presence of zero values, the set of 3000 proteins with the lowest amount of zero values were selected. For each sample, protein expression values were centered to a geometric mean of 10^7. From the Ivy Gap transcriptomics dataset, a total of 122 samples marked reference-histology from the same 5 niches were gathered, which included 25,872 genes. For each sample, transcriptomic data expressed as Transcript per Million (TPM) was centered to a geometric mean of 2.5.

Gene set signatures from the MSigDB-curated, MSigDB-hallmark, and MSigDB-oncogenic categories were retrieved, and 1410 signatures with at least 20 genes present in both the proteomics and transcriptomics datasets were selected. The ssGSEA scores were subsequently calculated for each signature, using the proteomics and transcriptomics datasets in parallel. For each signature, we computed a "signature coherence" ratio corresponding to the proportion of genes for which the signature's ssGSEA score and the expression levels of each gene show a linear model fit $p < 0.01$; then, we selected 985 gene signatures presenting a signature coherence ratio > 0.4 in both the proteomics and transcriptomics datasets. For each signature an XGBoost model that takes as input either proteomics or RNA expression data to infer the singature's ssGSEA scores, using the same set of empirically defined hyperparameters (eta = 0.1, subsample = 0.9, max.depth = 1).

To test the degree of RNA/protein concordance of each signature, we used the data from a third study comprised by 27 GBM samples that were profiled by proteomics and RNA-seq analysis, see Supplemental Source Data File (10.5281/zenodo.5593517)[51]. A signature was considered RNA/protein concordant when the predicted values produced by the XGBoost models for proteomics and transcriptomics data, when subjected to robust linear model fit (interquartile criterion for outlier sample removal), present a $p$ value<0.01. Here, ssGSEA was used to establish the ground truth in the training set, and the machine learning models were used to perform the actual concordance test in an external dataset due to their capacity to generalize patterns. The RNA/protein concordant analysis of the signatures resulted in the selection of a set of 64 signatures (Supplementary Data File 4). For our bioinformatical analysis throughout the study, we used R 4.0.3 and packages GSVA 1.40.1, xgboost 1.4.1.1, ComplexHeatmap 2.8.0, psSubpathway 0.1.1, Rtsne 0.15 and modelTsne (https://github.com/oicr-gsi/modelTsne).

**CT niche-specific inference of the gene signatures KRAS targets, MYC targets, and hypoxia.** Obtaining a clear picture of the biological status of the CT niche in terms of gene signatures activation, without interference from other tissue niches, can produce clinically relevant insights. Here, we developed a computational approach to infer the status of the gene signatures KRAS targets, MYC, and hypoxia, at the RNA level. Synthetic RNA expression data was generated by combining expression values from niche-specific samples (CT, PAN, LE, and MVP) from the Ivy-Gap study ($n = 98$), together with samples from the TCGA-LAML study under the category LEUKO ($n = 30$) to act as a surrogate for inflammatory infiltrate (Supplementary Fig. 35). During machine learning model training, the differences in the levels of inflammatory infiltrate that exist among the tissue niches lead to an excessive impact of the immune infiltrate-related variables in the "class definitions"; therefore, the inclusion of the LEUKO group is a strategy to exclude immune infiltrate-related variables from the non-LEUKO "class definitions" so that these are more focused on tissue-defining features. RNA-seq raw count data from niche-specific samples were transformed into "expanded counts", where a text vector of length equal to the total read count contains the name of each gene multiplied by its read count; then, a total of 10 million randomly sampled "expanded counts" were combined, where CT content must range between 0.2 and 0.8, and with PAN, LE, MVP and LEUKO completing the unit in random proportions; finally, the number of occurrences of each gene was computed, producing a numeric vector of gene counts spanning the transcriptome. The synthetic samples ($n = 202$) were split 80/20 into training and test subsets. For each signature, the first set of 11 "level 1" XGBoost models were trained to predict the ssGSEA score of the whole sample, the proportion of each niche (5 variables), and ssGSEA score from the original niche-specific sample (5 variables), respectively, using a set of empirically defined hyperparameters (eta = 0.2, subsample = 0.8, max.depth = 1). Next, an XGBoost model that takes as input the 11 "level 1" variables was trained to infer the CT niche-specific signature, which was defined as the ssGSEA score from the original CT niche-specific sample divided by the CT content present in the synthetic sample. Therefore, inference of CT niche-specific signature activation involves the prediction of the 11 "level 1" variables that subsequently are fed into the final XGBoost model (Supplementary Fig. 35). Kaplan-Meier survival group binning were based on "high" (equal to or higher than average) and "low" (below average) level of given signatures.

**Gene signature enrichment at the single-cell level.** Normalized scRNA expression values from glioblastoma samples, published by Richards et al., were downloaded from the Broad's Single Cell Portal (https://singlecell.broadinstitute.org/single_cell/study/SCP503). Using the expression data from the primary tumour samples (malignant cells only), cells with less than 10,000 UMIs were discarded to reduce signature enrichment bias due to low sequencing coverage. The ssGSEA scores for the signatures of interest (KRAS_targets, MYC_targets, HALLMARK_MITOTIC_-SPINDLE, and WU_CELL_MIGRATION) were then calculated at the cell level using the GSVA package[52]. Samples GBM_G1003-C (769 cells), GBM_G620 (261 cells),

and GBM_G983-C (434 cells) were selected as representative examples based on their abundance of KRAS_targets-high, MYC_targets-high, and of both cellular populations, respectively.

**RNA sequencing**. Fresh frozen tissue specimens were crushed by mortar and pestle, homogenized using the QIAShredder kit (Qiagen), and genomic DNA and total RNA were extracted using the AllPrep DNA/RNA Mini kit (Qiagen), according to the manufacturer's instructions. RNA libraries were synthesized using 200 ng of total RNA using the Illumina TruSeq Stranded RNA LT Sample Prep Kit (Illumina), and subsequently sequenced on the NextSeq550 platform to a read depth of 80 million clusters and 160 million paired-end reads (75 bp X 75 bp) using V2 chemistry. Finally, the resulting short reads were aligned to the human reference transcriptome (hg19) using STAR v2.6.0c, and RNA abundance was quantified with RSEM v1.3.0.

**Computational pharmacotranscriptomics screening**. To investigate whether the activation status of KRAS and MYC target genes and hypoxia is associated with differences in drug susceptibility in glioblastoma cell lines we retrieved the RNA-seq data (TPM format) from 33 glioblastoma cell lines from the Cancer Cell Line Encyclopedia (https://portals.broadinstitute.org/ccle). The level of activation of each signature was inferred using the RNA data from each cell line using the XGBoost models that were previously trained in the signature selection process. Drug sensitivity data of the dataset "CTRPv2" was obtained from the PharmacoGx package (https://bioconductor.org/packages/release/bioc/html/PharmacoGx.html). For each drug, those cell lines with available DSS1 values were ranked by the level of activation of the signature, then the bottom 25% and top 75% cell lines were assigned to the "low" and "high" activation groups, respectively. Differences in drug sensitivity between the two groups were considered significant if they met the following criteria: both groups possess at least of 4 members, one-way ANOVA's p-value <0.05, the difference in average DSS1 metric > 0.1, and also, average DSS1 < 0.1 in at least one of the groups. Finally, the results were visualized using the package EnhancedVolcano[53].

**Hypoxic tissue culture and chemical profiling**. To study the expression of AKAP12 and PTPRZ1 in-vitro, patient derived cultures were grown on 100 mm dishes (Corning Primaria, 353803), coated with poly-L-ornithine (Sigma, P4957) and laminin (Sigma, L2020). GSCs were grown in serum-free media containing EGF/FGF (base media: DMEM/F12, 1% antibiotic-antimycotic, 1% L-glutamine, N2 and B27). $1.0 \times 10^6$ cells were seeded and grown in 21% oxygen for a day prior to hypoxia treatment. They were exposed to hypoxia (0.2% Oxygen) and physioxia (5% Oxygen) in a H85 HypOxystation (Don Whitley Scientific) for 72 h before collecting cell lysates. For chemical profiling of kinase chemogenomic set (www.sgc-unc.org)[45], chemical probes (5 uM, 0.03% DMSO) were added to $5 \times 10^3$ cells were grown in a 96 well plate, 24 h after hypoxic incubation. On day 4, fresh media was supplemented with fresh media and chemical probe concentrations. Spheroids were incubated for an additional three days, at which cell viability was assessed by Alamar Blue Cell Viability Reagent (Thermo Fisher Scientific). All media and solutions were equilibrated in the appropriate hypoxia chambers overnight prior to use.

**Reporting summary**. Further information on research design is available in the Nature Research Reporting Summary linked to this article.

## Data availability
The anatomical atlas of human glioblastoma mass spectrometry proteomics data has been deposited to the ProteomeXchange Consortium via the PRIDE[50] partner repository with the dataset identifier PXD019381. The XGBoost mass spectrometry proteomics data have been deposited to the ProteomeXchange Consortium via the PRIDE[50] partner repository with the dataset identifier PXD029290. The RNA data generated in this study have been deposited in the Zenodo database under accession code https://doi.org/10.5281/zenodo.5639569[51]. The RNA-seq dataset was deposited in the European Genome-Phenome Archive with the accession identifier EGAD00001008362. Data used in this publication were generated by the Clinical Proteomic Tumor Analysis Consortium (NCI/NIH) and are publicly available [https://cptac-data-portal.georgetown.edu/study-summary/S048]. Data used in this publication were generated by The Cancer Genome Atlas Program (TCGA) and deposited at the Data Coordinating Center (DCC) for public access [http://cancergenome.nih.gov/]. The IvyGAP data including the RNA-Seq and copy number data are publically available at Gene Expression Omnibus through GEO series accession number GSE107560. The single-cell are publicly available through the Broad Institute Single-Cell Portal (https://singlecell.broadinstitute.org/single_cell/study/SCP503) and CReSCENT60 (https://crescent.cloud; study ID CRES-P23). The data generated in this study are provided in the Supplementary Information/Source Data file. RNA-seq data and drug sensitivities from 33 glioblastoma cell lines from the Cancer Cell Line Encyclopedia are accessible at (https://portals.broadinstitute.org/ccle). The MSigDB database can be accessed at (https://www.gsea-msigdb.org/gsea/index.jsp). Source data are provided with this paper.

## Code availability
Activation and analysis of the 64 gene signatures can be found at (https://github.com/diamandis-lab/paper-prot-atlas-gbm). The activation of these signatures can also be found at 10.5281/zenodo.5576255. It can also be accessed via an online tool (https://cancerhub.shinyapps.io/prot-atlas-gbm/).

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

## Acknowledgements

We would like to thank the Structural Genomics Consortium (SGC) for providing the chemical probes for our study. The Diamandis Lab is supported by the Canadian Cancer Society, Canadian Research Society, the Terry Fox New Investigator Award program, the Canadian Institute of Health Research, and the Princess Margaret Cancer Foundation. UD is supported by the Richard Motyka Brain Tumour Research fellowship of the Brain Tumour Foundation of Canada. AK was supported by the Hungarian National Brain Research Program (NAP 2.0, 2017-1.2.1-NKP-2017-00002). Ontario Institute for Cancer Research (OICR) provided funding for the RNA sequencing

## Author contributions

B.L., U.D., and P.D. conceived of the project. P.D. and M.R. selected and annotated the tumor samples. B.L. carried out the LCM, sample preparation, and MS analysis. S.L., B.L. conducted hypoxia in-vitro work. W.H. conducted the chemical screen. B.L. and A.L. performed the bioinformatic analysis. B.L. and K.F. created the online resource. A.K., G.H., provided the glioblastoma samples for the proteogenomic XGBoost model. P.D., U.D., and B.L. interpreted the data and wrote the manuscript with input from all other authors.

## Competing interests

The authors declare no competing interests.
