## [Peer review file · Nature Communications]

REVIEWER COMMENTS

Reviewer #1 (Remarks to the Author): Expert in glioblastoma

Overall, this paper provided interesting molecular profile of intra-tumoral heterogeneity by regionally isolated and defined tissue proteomics. This approach is unique and powerful to show the relationship between tumor microenvironment (TME) and tumor progression. Moreover, by using machine learning method, they also demonstrated new topographic map. This map demonstrates that MYC- and KRAS-axis cooperate with hypoxia to produce three-dimensional model for intra-tumoral heterogeneity and provide clinical relevance for chemoresistance in GBM. Although, their approaches are novel and interesting, they did not provide enough evidence to strongly support their conclusions. Moreover, there is not much progress that provided by proteomics results compared with previous transcriptomic analysis.

Major points

1. In this paper, authors collected different anatomical niches by laser capture microscopy (LCM) to excise regions of IT, CT, PAN, and MVP by benchmarking IVY GAP. However, these samples might have different TME proportions and variable tumor purity. How did they normalize these differences among patient samples?

2. By proteomic analysis from FFPE samples, authors identified total 4,794 proteins across the entire sample cohort. However, heterogeneity of tissue composition might contribute to loss of common proteins that were identified and quantified entire set of proteomic analysis. How many proteins were commonly identified and quantified entire samples and how did they quantify missing values? Among different anatomical niches, identified number of proteins are extremely variable. Can they compare statistic power between proteomics and transcriptomics?

3. In Fig 3H, authors validate selective localization of CD276 to MVP region and argued the potential contribution of immunosuppressive GBM microenvironment. They also suggest possible application of anti-CD276 antibody as a immune check point inhibitor. Can they provide any other evidence that can support possible immune environment for this treatment in this region? Are there any other proteomic and transcriptomic expressions related to immune signatures?

4. To extract functionally relevant gene sets at the individual sample level, independent of regional groups, they did hierarchical clustering of purified tumor tissue regions (CT and PAN samples). Is there any reason why they exclude other regions? How they control contamination or purity issue among different samples?

5. In Fig 4, authors found KRAS signature and MYC target signature were significantly enriched in

Cluster 1 and Cluster 2. These clusters were similar to Verhaak's Mesenchymal and Proneural subclasses. When were you comparing your gene signature analysis between your clusters and Verhaak's subclasses, how different they are and what is strength in your clustering?

6. In Fig 4D, and E, authors included third axis as "Hypoxia" molecular signature. However, hypoxia signature seems to be more associated PAN regions than CT regions. Is there any clear reason why hypoxia signature is most important third axis compare to other signatures?

7. In fig 6, authors demonstrated pharmacological profiling of axis-specific drug vulnerabilities and resistance. They show that KRAS-signature show more resistance to MYC-signature. However, cellines were grown in the same condition without any consideration about tumor microenvironment. How can you directly compare KRAS- and MYC- signature in the same culture condition?

8. Authors identified key signaling axes, KRAS- and MYC-signatures, that can potentially pharmacologically profile GBM patients. Can they also apply this signature to single cell analysis datasets that was already available in other cohorts?

Minor points

1. In Fig 4D, and E, description on figures is different from the description in legend and manuscript.

2. In Fig 5E, and F, description on figures is different from the description in legend and manuscript.

Reviewer #2 (Remarks to the Author): Expert in proteomics, pharmacogenomics, and systems biology

The paper by K.H. Brian Lam et al. addresses intra-tumor heterogeneity of histomorphologic regions across 20 patient samples of glioblastoma. They approached this using a mass spectrometry based quantitative proteomics approach and additional mining of public transcriptomic and proteomic datasets. They largely adopted the concept of the IVYGAP study (Puchalski et al. Science 2018) in which transcriptomic level characterization was conducted for the same anatomic features (LE, IT, CT, PAN, and MVP). These authors further refined the associations by intersecting their quantitative proteomic data.

The main finding of this study is the identification of two distinct neoplastic cell classes, MYC- and KRAS-targets activated classes, wherein hypoxia signature serves as an orthogonal axis with which to further distinguish each class to PAN and CT. Using machine learning regression model-based inference of CT-niche specific signatures by mining public bulk RNA-seq data, they showed patients with an enhanced KRAS-signature exhibit poor prognosis. Likewise, by stratifying GBM cell lines in the public pharmacogenomics dataset into KRAS- and MYC-enriched subgroups, they found that the KRAS-subgroup displays chemoresistance.

While the manuscript is interesting and clearly patient relevant, the study novelty is somewhat incremental, and the conclusions are not yet definitive. Below are some suggestions for the authors to consider to further improve their manuscript.

Major comments:

1. Due to the limited sample size, binary classification of glioblastoma neoplastic cells into KRAS- or MYC-activated class is unclear. IDH wildtype GBMs are typically classified into three (or four) transcriptomic subtypes, including proneural, classical, and mesenchymal subtypes. The IVYGAP paper (Puchalski et al. Science 2018) claimed via inference that CT and PAN contributes the majority of the bulk RNA-seq outcomes (Fig S8 of IVY GAP paper). This suggest that, if there is sufficient concordance between mRNA and protein levels, the authors might expect at least three (proneural-like, classical-like, mesenchymal-like) proteomic subtypes. Why did the authors's main signatures not include the classical-like subtype? Is this just a sample size issue (number of CT and PAN = 34), a lack of mRNA and protein correlation for genes determining the classical subtype, or overfitting of the prior classification?
2. Unless the authors have clear experimental evidence, naming the two subgroups as MYC- or KRAS-target enriched subgroups, could mislead people, since KRAS mutation is not commonly found in GBM. Instead, proneural-like and mesenchymal-like may better represent the two subtypes.
3. The authors assessed the clinical utility of the two subtypes two-fold: estimating differences in prognosis and deriving subtype-specific treatments. They claimed that the KRAS class has worse prognosis based on a niche-specific inference of the TCGA GBM dataset. However, the mesenchymal subtype of GBM has been shown to have worse prognosis than other transcriptome subtypes (Wang et al. Cancer Cell 2017). As their KRAS-target enriched class is associated with GBM mesenchymal features, it is not surprising that they show worse prognosis. Pharmacological analysis of the two major classes is limited to computational analysis of public data originally obtained in 2D culture conditions with physiologically discordant GBM cell line models. The authors should demonstrate at least one novel clinical utility of the classification to be suitable for Nature Communications.
4. Pharmacological profiling of GSC cells against 188 kinase inhibitors is somewhat off the table. It would be more relevant if the proteomic class (KRAS or MYC) for GSC cells is determined first and show whether any of drugs have selective cytotoxicity in the physiologically relevant condition (hypoxia). However, the results indicated that none of the tested drugs had cytotoxicity in hypoxic conditions.
5. Methods how the authors identified the hypoxia axis is not described in detail. Also, the biological and clinical importance of the axis is not sufficiently explored in the manuscript, except for the abovementioned drug screening data with the GSC model. To claim this as a main axis, further evidence might be needed to support its significance.
6. Justification of including the "LEUKO" dataset for the niche-specific gene signature inference should be addressed in more detail. Stating simply "to act as a surrogate for inflammatory infiltrate" in the Methods is insufficient, since some anatomical regions, particularly MVP, might include signals from immune infiltrates, which might affect the accuracy of the ML models.

Minor comments:

1. In Fig. 2C, it is hard to discern point colors. Use more clear color scales.
2. In Fig. 4B, Cluster 1 and Cluster 2 labels should be moved to columns.
3. 3D plots in Fig. 4D and 4E are very confusing. Please use a different display method, such as 2D + color code.
4. Fig. 4D and 4E seem to be switched or mislabeled.
5. In Fig. 5C, "IvyGap" is underlined in red. Fix this.
6. Fig. 5E and 5F seem to be switched or mislabeled.
7. In Fig. 5E and 5F, please describe how the patients were divided into two groups.
8. Enlarge font sizes for labels in Fig. 6D and other figures.
9. In page 7, line #10, "clinical outcomes of glioma patients from TCGA." Is this glioma or GBM?
10. Software versions and database versions are missing in the Methods section.
11. Cannot find files or links for computer codes and R scripts used to generate the results.

Reviewer #3 (Remarks to the Author): Expert in glioblastoma genomics

This study represents an important resource for the community and the authors should be thanked for performing regionally-resolved proteomic analysis in GBM and making the data available to the community. I have minor comments.

- colors in figure 1C are very hard to distinguish; a different color scheme should be used;
- figure 3F is overstained;
- there are numerous survival analyses throughout the study and it is sometimes unclear what the message is; in the first part of the study, the analysis is focused on defining regionally-resolved protein-expression programs; but then typically only 1 protein is shown/chosen to correlate with survival; did other genes in e.g. PAN (other than AKAP12) also correlate with survival? (i.e. is it the abundance of PAN that correlated or is this specific to AKAP12); a similar question would be relevant for CD276 and MVP; is it MVP-globally or CD276-specific? It also seems that CD276 is quite highly expressed in PAN and it would be important to show the staining pattern in such areas as well?
- figure 5F: the survival analysis seems to not replicate so well in CPTAC dataset; would be important to explain discrepancies/again downplay clinical relevance if needed;
- the title/chosen nomenclature around KRAS and MYC is a somewhat simplification; other programs are part of those signatures (e.g. KRAS with mesenchymal and MYC with proneural); so one could also chose to use this "more established" GBM nomenclature; a rationale for focusing on KRAS and MYC naming/subprograms should be provided and the survival analyses should be done with (or compared to) the entire programs, not subprograms;
- In general, this reviewer feels that showing correlation to survival is ok, but many many markers have been suggested in GBM and tend to end up not being that useful; so a resource-focused paper, rather than too many survival curves would be preferable;

Point by Point response to reviewer comments

We thank reviewers for your efforts, time and valuable feedback during the challenging times with COVID-19 and hope you and your families are keeping safe. We have now had a chance to review your comments and make all the recommended changes including expanding our analytical interruptions of our triple axis model. We also include additional analyses to highlight clinical utility of our findings and carrying out the additional recommended analyses.

Specifically, the revised manuscript addresses, among others, the following major three aspects raised: (i) quantification of immune content (using CIBERSORT) to characterize the immune differences across tissue niches, (ii) addition of single-cell RNA expression that provides further support of our triple axis model, and (iii) justification of the use of the gene signatures of our model (KRAS_TARGETS, MYC_TARGETS and HYPOXIA) instead of the more commonly used classification system by Verhaak (classical, mesenchymal and proneural).

Please find point-by-point responses to your comments and suggestions below. We believe your insights have substantially improved the quality of our work and we thank you for this constructive feedback.

Reviewer 1:

Overall, this paper provided interesting molecular profile of intra-tumoral heterogeneity by regionally isolated and defined tissue proteomics. This approach is unique and powerful to show the relationship between tumor microenvironment (TME) and tumor progression. Moreover, by using machine learning method, they also demonstrated new topographic map. This map demonstrates that MYC- and KRAS-axis cooperate with hypoxia to produce three-dimensional model for intra-tumoral heterogeneity and provide clinical relevance for chemoresistance in GBM. Although, their approaches are novel and interesting, they did not provide enough evidence to strongly support their conclusions. Moreover, there is not much progress that provided by proteomics results compared with previous transcriptomic analysis. *The submission could be significantly strengthened by considering the following changes:*

Comment R1.1: *“In this paper, authors collected different anatomical niches by laser capture microscopy (LCM) to excise regions of IT, CT, PAN, and MVP by benchmarking IVY GAP. However, these samples might have different TME proportions and variable tumor purity. How did they normalize these differences among patient samples?”*

Response R1.1: Thank you for your interest in our work and your comments for improvements. We used specific selection criteria for each anatomical niche based on hallmark histomorphologic features comparable to the existing IVY GAP GBM Atlas strategy. We now add the criteria used explicitly for clarity (Methods, page 17, Lines 9-19). To ensure this would be a valuable study and resource, we reviewed over one hundred cases and selected only those in which at least 3 regions were well represented and had the classic and immediately recognizable histologic patterns. These were reviewed independently by two board certified neuropathologists and the cases were included only upon their unanimous agreement. To further increase transparency of each patient sample, we now include representative H&E images before and after LCM and present their respective spatial resolution within a PCA plot (Supplemental figures 2-25).

New Text in the Methods (Page 15, Lines 9-19): “TME differences across patient samples were normalized using the same selection criteria as IvyGAP. Leading Edge (LE) is the outermost boundary of the tumor, where the ratio of tumor to normal cells is about 1-3/100, and the laminar architecture of the cortical layers is frequently evident. Infiltrating Tumor (IT) is the intermediate zone between the Leading Edge (LE) and Cellular Tumor (CT), and is frequently marked by perineuronal satellitosis. Cellular

Tumor (CT) constitutes the major part of the tumor core, where the ratio of tumor cells to normal cells is between 100/1 to 500/1. Pseudopalisading Cells around Necrosis (PAN) is the narrow boundary of cells arranged like pseudopalisades along the perimeter of necrosis. Microvascular Proliferation (MVP) refers to two or more blood vessels sharing a common vessel wall of endothelial and smooth muscle cells arranged in the shape of a glomerulus or garland of multiple interconnected blood vessels.”

Comment R1.2: “By proteomic analysis from FFPE samples, authors identified total 4,794 proteins across the entire sample cohort. However, heterogeneity of tissue composition might contribute to loss of common proteins that were identified and quantified entire set of proteomic analysis. How many proteins were commonly identified and quantified entire samples and how did they quantify missing values? Among different anatomical niches, identified number of proteins are extremely variable. Can they compare statistic power between proteomics and transcriptomics?”

Response R1.2: We acknowledge that biological heterogeneity in tissue composition is expected to lead to differences in identified and quantified proteins. To address this comment, we now include numbers of commonly identified proteins within each anatomical niche in a supplementary figure (See Supplementary Fig. 1) (1330 proteins within all CT samples, 1316 within all IT samples, 947 proteins within all LE samples, 1182 proteins within all MVP samples and 1303 proteins within all PAN samples). For the missing values, we utilize an imputation algorithm by replacing them along a defined standard distribution of random values, standard practice in MS-based proteomic studies, for downstream statistical analysis (See Page 20, Line 17-19). For proteogenomic concordance, we perform spearman rank correlation analysis between our data set and IvyGAP (Fig. 2A) and identify a number of proteogenomic concordant enriched genes within each anatomical niche (Fig. 2C).

New Text in the Methods (See Page 20, Line 17-19): “Proteins were filtered such that only those that appeared in at least 60% within a group were included. The raw values were Log₂ transformed and non-valid values were imputed (downshift=0.3, width=1.8).”

Comment R1.3: In Fig 3H, authors validate selective localization of CD276 to MVP region and argued the potential contribution of immunosuppressive GBM microenvironment. They also suggest possible application of anti-CD276 antibody as a immune check point inhibitor. Can they provide any other evidence that can support possible immune environment for this treatment in this region? Are there any other proteomic and transcriptomic expressions related to immune signatures?

Response R1.3: To address your question, we now apply a digital cytometry technique called CIBERSORT (Newman et al, Nature Methods, 2015 & Newman et al, Nature Biotechnology, 2019) to characterize immune cell compositions from the proteomic profiles of each region. Using this approach, we show a relative depletion of memory CD4 T cells in both the MVP and PAN compartments. We observed a heterogenous distributioin of T cells in one of our recent studies on T infiltrating lymphocytes in GBM (Alfsafwani et al, JNEN, 2021). This is also supportive of the role of CD276 in suppressing T helper cell immune response in other models (WK Suh, Nature Immunology, 2003). We now include this new analysis of immune cell infiltrates and relevant literature in the revised figures (Fig. 3j-k and Supplementary Fig. 28).

We also highlight that the enrichment of CD276 within vasculature of other cancers and note anti-CD276 therapies (NIH) (Please see Page 14 lines 3-15). To identify proteomic immune signatures we perform GSEA to identify immune processes enriched in MVP. We now highlight a number of pathways including allograft rejection and neutrophil activity in the modified figure (Please see Fig. 2e).

Comment R1.4: To extract functionally relevant gene sets at the individual sample level, independent of regional groups, they did hierarchical clustering of purified tumor tissue regions (CT and PAN samples). Is there any reason why they exclude other regions? How they control contamination or purity issue among different samples?

Response R1.4: Thank you for your comments. For clarification, we chose to focus on only CT and PAN for downstream analysis, as these regions contained the highest proportion of pure tumor cells (nearly 100% by histology). Conversely, given their definition, MVP, IT, and LE samples contain a significant proportion of non-neoplastic cellular elements. By utilizing only CT and PAN samples for downstream analysis, we believe we could sufficiently overcome contamination of the most relevant non-neoplastic cell types (e.g. endothelial cells, normal brain tissue elements). For sample purity, we utilized pathologist-defined regions as defined by the IvyGAP study (Please see Supplementary Figs. 2-25). As mentioned above, we screened a large number of cases to control for niche variations across patient samples and created the presented high-quality cohort for our regional analysis.

New text included (Page 8, Lines 21-24): “For this analysis, we chose to focus on only CT and PAN, as these regions contained the highest proportion of pure tumor cells (nearly 100% by histology) compared to other regions like MVP, IT, and LE that contained a significant proportion of non-neoplastic cellular elements that could confound the analysis.”

Comment R1.5: In Fig 4, authors found KRAS signature and MYC target signature were significantly enriched in Cluster 1 and Cluster 2. These clusters were similar to Verhaak’s Mesenchymal and Proneural subclasses. When were you comparing your gene signature analysis between your clusters and Verhaak’s subclasses, how different they are and what is strength in your clustering?

Response R1.5: As suggested, we now provide a comparative analysis of the different clustering results with the 64 selected signatures, the KRAS_TARGETS + MYC_TARGETS signatures, and Mesenchymal + Proneural (Supplementary Fig. 30). We now provide a more detailed explanation behind our rationale for building a GBM disease model based on the expression status of the KRAS and MYC targets instead of using the GBM classification system proposed by Verhaak (See Fig. 4G). This receiver operator curve (ROC) analysis of our samples using either the Verhaak signature sets (0.91 and 0.82 for mesenchymal and proneural, respectively) and our MYC and KRAS signatures (0.95 and 0.88 respectively) reveals better performance of our model in classifying our samples into distinct clusters. Furthermore, we compare p-value rankings of different MsigDB modules for both of the Verhaak gene sets and those of KRAS and MYC demonstrating improved compactness of the molecular modules of our proposed axes (Supplementary Fig. 31). This latter feature turned out to be quite important when comparing drug sensitivity differences of 31 GBM cell lines later in the paper as much fewer axis-specific agents were identified using the traditional transcriptionally-defined subgroups. These findings are described in the modified manuscript as follows:

New text included (Page 9, Lines 14-23): “Of note, the Verhaak_GBM_classical signature did not appear in our set of 64 proteogenomic signatures due to low protein/RNA concordance (robust linear fit’s $p=0.059$). Notably, both this signature and the neural subgroup (which did have good protein-RNA correlation) did not meaningfully correlate with the two defined protein-based clusters. Given the aforementioned proteogenomic discordances in cancer programs and contextual differences (e.g. bulk RNA profiling), we used the area under the receiver operator curve (AUC, ROC) to define the nomenclature of our spatially-profiled and protein clusters (**Fig. 4g**). Importantly, in our proteomic datasets, MYC and KRAS gene set-driven clustering yields more robust sample separation than Verhaak Mesenchymal and Proneural gene sets (**Supplementary Fig. 30**).”

Comment R1.6: In Fig 4D, and E, authors included third axis as “Hypoxia” molecular signature. However, hypoxia signature seems to be more associated PAN regions than CT regions. Is there any clear reason why hypoxia signature is most important third axis compare to other signatures?

Response R1.6: We appreciate this comment regarding the third axis of our model. For clarification, our unsupervised analysis yielded two major clusters/axes (MYC and KRAS-related clusters). The intermixing of PAN and CT however was somewhat surprising to us given the visible histopathologic pattern differences of these regions. As a quality control and additional separation layer, we therefore wanted to see how the hypoxia signature further segregates the two defined clusters. Remarkably, PAN regions could be reliably differentiated from CT regions using this signature (See ROC analysis in Fig. 4g). We therefore felt this was an important third axis to incorporate into our model because it provided a strong account for the known hypoxic biology presumed to exist in PAN regions and highlighted the robustness of our niche selection and analytical pipeline. We now better clarify the motivation for this third axis in the revised manuscript as follows:

New text included (Page 9, Lines 5-13). “Closer inspection of signature enrichment patterns revealed an inverse correlation between the KRAS targets and MYC targets signatures (Fig. 4c), and, despite being intermixed between CT and PAN regions, these anatomical coordinates appeared to be relatively further faithfully segregated across a third “hypoxia” molecular signature axis that was also one of the concordant proteogenomic programs defined by our analysis (Fig. 4d). This triple axis of separation was validated by interrogation of the independent IVY gap transcriptional atlas (Fig. 4e), and confirming, within the CT samples, that high KRAS targets activity is associated with invasion and epithelial-to-mesenchymal transition processes whereas samples enriched for the MYC axis were associated with cell cycle progression (Fig. 4f).”

Comment R1.7: In fig 6, authors demonstrated pharmacological profiling of axis-specific drug vulnerabilities and resistance. They show that KRAS-signature show more resistance to MYC-signature. However, cell lines were grown in the same condition without any consideration about tumor microenvironment. How can you directly compare KRAS- and MYC- signature in the same culture condition?

Response R1.7: We appreciate the comment. We note that both the KRAS and MYC signatures were derived from highly pure tumor regions and not necessarily dependant on specific extrinsic tumor microenvironments. The pharmacoproteomic analysis was used to highlight the potential significance of the model we proposed. This was done by exploring transcriptional patterns of 33 different GBMs in the CCLE resource (Broad Institute) and ranking their expression profiles along the different axes. This allowed us to compare influence of KRAS- and MYC-signatures across multiple GBMs in the same culture condition.

Comment R1.8: Authors identified key signaling axes, KRAS- and MYC-signatures, that can potentially pharmacologically profile GBM patients. Can they also apply this signature to single cell analysis datasets that was already available in other cohorts?

Response R1.8: To address your comment, we now explore our proposed axes in single cell data from a recent publication as a complementary validation analysis (Richards et al., Nature Cancer 2021). This data is now included (Fig. 4h-j and Supplementary Fig. 33). Specifically, we show that transcriptional single cell profiles of GBMs show variable expression profiles that can be scored along the KRAS and MYC enrichment axis and that these axes are mutually exclusive to one another. Moreover, the major biological processes we found to be enriched along these 2 axes are also concordant in this single cell

analysis, further strengthening our proposed model. Lastly, this new data further supports and validates that the axis signals are driven from tumor cell programs and not contaminated from other non-neoplastic TME components in the regions. This new analysis is now described in the text.

New text included (Page 10, Lines 6-18): “To further characterize intra-tumour variability, the enrichment level of the KRAS_TARGETS and MYC_TARGETS signatures was assessed in a group of samples at the single-cell level by using scRNA data from GBM cells isolated from bulk tumour samples⁴⁴. The GBM cells were grouped as KRAS_TARGETS-high, MYC_TARGETS-high, together with a “Central group” that shows no elevated enrichment in either signature (**Fig. 4h** and **Supplementary Fig. 33**). Cells highly enriched for both signatures are nearly non-existent, supporting the observed mutually exclusive feature of these cellular phenotypes. The MYC_TARGETS-high group presented high enrichment of the HALLMARK_MITOTIC_SPINDLE signature (**Fig. 4i**) whereas KRAS_TARGETS-high cells were enriched in WU_CELL_MIGRATION, a signature that is associated with tumour invasiveness (**Fig. 4j**). These results support a model where a large proportion of GBM cells within a tumour tissue appear to be “oncogenetically stable”, with certain subpopulations exhibiting “oncogenic activation” towards either invasive or proliferative phenotypes.”

Minor Comments:

Comment R1.9: In Fig 4D, and E, description on figures is different from the description in legend and manuscript. In Fig 5E, and F, description on figures is different from the description in legend and manuscript.

Response R1.9: The figures have been corrected accordingly in the revised version.

Reviewer 2:

The paper by K.H. Brian Lam et al. addresses intra-tumor heterogeneity of histomorphologic regions across 20 patient samples of glioblastoma. They approached this using a mass spectrometry based quantitative proteomics approach and additional mining of public transcriptomic and proteomic datasets. They largely adopted the concept of the IVYGAP study (Puchalski et al. Science 2018) in which transcriptomic level characterization was conducted for the same anatomic features (LE, IT, CT, PAN, and MVP). These authors further refined the associations by intersecting their quantitative proteomic data.

The main finding of this study is the identification of two distinct neoplastic cell classes, MYC- and KRAS-targets activated classes, wherein hypoxia signature serves as an orthogonal axis with which to further distinguish each class to PAN and CT. Using machine learning regression model-based inference of CT-niche specific signatures by mining public bulk RNA-seq data, they showed patients with an enhanced KRAS-signature exhibit poor prognosis. Likewise, by stratifying GBM cell lines in the public pharmacogenomics dataset into KRAS- and MYC-enriched subgroups, they found that the KRAS-subgroup displays chemoresistance.

While the manuscript is interesting and clearly patient relevant, the study novelty is somewhat incremental, and the conclusions are not yet definitive. Below are some suggestions for the authors to consider to further improve their manuscript.

Comment R2.1: Due to the limited sample size, binary classification of glioblastoma neoplastic cells into KRAS- or MYC-activated class is unclear. IDH wildtype GBMs are typically classified into three (or four) transcriptomic subtypes, including proneural, classical, and mesenchymal subtypes. The IVYGAP paper (Puchalski et al. Science 2018) claimed via inference that CT and PAN contributes the majority of the bulk RNA-seq outcomes (Fig S8 of IVY GAP paper). This suggest that, if there is sufficient concordance between mRNA and protein levels, the authors might expect at least three (proneural-like, classical-like, mesenchymal-like) proteomic subtypes. Why did the authors's main signatures not include the classical-like subtype? Is this just a sample size issue (number of CT and PAN = 34), a lack of mRNA and protein correlation for genes determining the classical subtype, or overfitting of the prior classification?

Response R2.1: Thank you for raising this important point. In short, the classic-like signature was excluded due to a lack of significant mRNA to protein correlation. Our final cohort, while relatively small, was generated from a much larger number of cases after carefully selecting for relatively large GBM resections in which the majority of niches could be objectively identified and sufficiently isolated in a meaningful manner. By using more than 5,000 gene signatures of the MSigDB as the starting point, we carried out a computational process to select those that are informative in GBM at the protein level. This was largely driven by identifying gene sets with strong protein/RNA concordance. This led to a data-driven nomination of 64 signatures, which were then applied to our dataset. From the 4 Verhaak classes, the Proneural, Mesenchymal and Neural classes met all the selection criteria and were part of the set of 64 signatures. As you know, the relevance of the neural subgroup has recently been challenged, and in our analysis, also did not appear to be specific to either of our two defined clusters. The Verhaak classical signature was excluded due to poor protein/RNA concordance (robust linear fit's $p=0.059$). We now include a statement in the manuscript providing our reasoning for not including the classical signature (around page 9, lines 14-15). For completeness, we also include an ROC analysis for the classical signature to also show it did not meaningfully correlate with the defined proteomic clusters (Fig 4G and also see Supplemental Fig. S30).

Comment R2.2: Unless the authors have clear experimental evidence, naming the two subgroups as MYC- or KRAS-target enriched subgroups, could mislead people, since KRAS mutation is not commonly found in GBM. Instead, proneural-like and mesenchymal-like may better represent the two subtypes.

Response R2.2: We appreciate this comment to increase the clarity of our work. We agree that KRAS mutations are rarely found in GBM. This axis however does not imply mutational dysregulation and it is important to point out that multi-dimensional genomic analyses from TCGA have pinpointed “deregulation of the RTK/RAS/PI(3)K pathway as an obligatory event in most or perhaps all glioblastomas” (McLendon, R. et al. Comprehensive genomic characterization defines human glioblastoma genes and core pathways. Nature 455, 1061–1068, 2008). Based on our newly included ROC analysis, MYC and KRAS-target axis better distinguish the identified clusters (Fig. 4g and Supplemental Fig. S30), at least at the proteomic level. We further clarified the statistical and contextual motivation for choosing the proposed nomenclature over those found in bulk-based genomic studies in the response to Reviewer 1, Comment R1.5 and new section in the text below.

New text included (Page 9, Lines 14-30): “Of note, the Verhaak_GBM_classical signature did not appear in our set of 64 proteogenomic signatures due to low protein/RNA concordance (robust linear fit's $p=0.059$). Notably, both this signature and the neural subgroup (which did have good protein-RNA correlation) did not meaningfully correlate with the two defined protein-

based clusters. Given the aforementioned proteogenomic discordances in cancer programs and contextual differences (e.g. bulk RNA profiling), we used the area under the receiver operator curve (AUC, ROC) to define the nomenclature of our spatially-profiled and protein clusters (**Fig. 4g**). Importantly, in our proteomic datasets, MYC and KRAS gene set-driven clustering yields more robust sample separation than Verhaak Mesenchymal and Proneural gene sets (**Supplementary Fig. 30**).”

Comment R2.3: The authors assessed the clinical utility of the two subtypes two-fold: estimating differences in prognosis and deriving subtype-specific treatments. They claimed that the KRAS class has worse prognosis based on a niche-specific inference of the TCGA GBM dataset. However, the mesenchymal subtype of GBM has been shown to have worse prognosis than other transcriptome subtypes (Wang et al. Cancer Cell 2017). As their KRAS-target enriched class is associated with GBM mesenchymal features, it is not surprising that they show worse prognosis. Pharmacological analysis of the two major classes is limited to computational analysis of public data originally obtained in 2D culture conditions with physiologically discordant GBM cell line models. The authors should demonstrate at least one novel clinical utility of the classification to be suitable for Nature Communications.

Response R2.3: We agree that other gene expression signatures have been proposed to predict disease outcome in GBM (including the mesenchymal signature). We note however the mentioned gene signature is quite large and span multiple gene modules and a large array of genes that make its utility potentially less valuable for clinical implementation and therapeutic applications. The KRAS signature we propose, in addition to better representing our generated proteomic dataset, is also more compact (**Supplementary Supplementary Fig. 31**). We believe showing a survival difference with a “cleaner” signature can therefore be seen as progress in classification as it may be subject to less overfitting than the mesenchymal signature.

We now further discuss important differences between KRAS and mesenchymal signatures and believe they provide insights into defining more refined phenotypic models of how glioblastoma heterogeneity could be conceptualized (See Response to Reviewer 1, Comment R1.5).

Despite the stated limitations of our pharmacological analysis, we feel it provides relevant information of how these axes may be contributing to resistance. As clinical management of cancer progresses towards more specific and precise molecular therapies, having a refined model of inter- and intra-tumoral heterogeneity of GBM at the proteomic level is of biological relevance with future clinical utility. To further demonstrate the clinical utility of these refined signatures using the existing pharmacogenomic signatures, we show that doing the same analysis with these potentially less specific mesenchymal and proneural signatures leads to fewer significant pharmacological differences across these different molecular axes. We believe highlight the superior pharmacological sensitivity predictive power of our molecular programs to be a relevant innovation for the GBM field.

Comment R2.4: Pharmacological profiling of GSC cells against 188 kinase inhibitors is somewhat off the table. It would be more relevant if the proteomic class (KRAS or MYC) for GSC cells is determined first and show whether any of drugs have selective cytotoxicity in the physiologically relevant condition (hypoxia). However, the results indicated that none of the tested drugs had cytotoxicity in hypoxic conditions.

Response R2.4: Thank you for your comment. We carried out pharmacological profiling under “hypoxic” vs “normoxic” in order to complement the pharmacogenomic analysis along the KRAS and MYC axis. We believe our experiment does indeed show that our third proposed axis also has important therapeutic implications due to altered drug sensitivity differences. To answer your second question, we

did discover some drugs that possessed more significant cytotoxicity in hypoxia than in normoxia. We now add some images for one such drug (GSK949675) that inhibits AKT1, AKT2, and PRKCH in Supplementary Fig. 36 to better illustrate this. Together with the MYC-KRAS pharmacological analysis, this suggests that all three axes can affect drug sensitivity profiles of GBM cells which was the insight we wanted to emphasize with these experiments. We believe that, in combination with the signatures derived from clinical patient samples, this improves potential clinical implications from existing models (e.g. transcriptional subgroups) and available datasets (e.g. chemical profiling of GBM cells in a single oxygen concentration). We discuss opportunities to carry out more complex screens simultaneously across all three axes (MYC, KRAS and Hypoxia) within a single experiment in the discussion as an exciting future direction (See Page 14-15, Lines 16-16).

Comment R2.5: Methods how the authors identified the hypoxia axis is not described in detail. Also, the biological and clinical importance of the axis is not sufficiently explored in the manuscript, except for the abovementioned drug screening data with the GSC model. To claim this as a main axis, further evidence might be needed to support its significance.

Response R2.5: We appreciate the omission of these details in the original manuscript. We have now improved our description and motivation for this axis, specifically the strong ROC performance at resolving PAN and CT regions in our study (See response to comments R1.5, R2.3 and R2.4).

Comment R2.6: Justification of including the “LEUKO” dataset for the niche-specific gene signature inference should be addressed in more detail. Stating simply “to act as a surrogate for inflammatory infiltrate” in the Methods is insufficient, since some anatomical regions, particularly MVP, might include signals from immune infiltrates, which might affect the accuracy of the ML models.

Response R2.6: To address this concern, the following has been added to the manuscript:

New included text (Page 22-23, Lines 28-2): “together with samples from the TCGA-LAML study under the category LEUKO (n=30) to act as surrogate for inflammatory infiltrate. During machine learning model training, the differences in the levels of inflammatory infiltrates that exist among the tissue niches lead to an excessive impact of the immune infiltrate-related variables in the “class definitions”; therefore, the inclusion of the LEUKO group is a strategy to exclude immune infiltrate-related variables from the non-LEUKO “class definitions” so that these are more “focused” on tissue-defining features.”

Minor Comments

Comment R2.7: In Fig. 2C, it is hard to discern point colors. Use more clear color scales.

Response R2.7: Thank you for your suggestion, we have now changed the colour scheme

Comment R2.8: In Fig. 4B, Cluster 1 and Cluster 2 labels should be moved to columns.

Response R2.8: Fig. 4b has been reformatted.

Comment R2.9: 3D plots in Fig. 4D and 4E are very confusing. Please use a different display method, such as 2D + color code.

Response R2.9: We agree that the interpretation of the 3D plots may require a slight effort from some readers. We have explored alternatives, both prior to the initial submission and during the revisions process, but have not found a visualization method that better captures the essence of our model.

Comment R2.10: Fig. 4D and 4E seem to be switched or mislabeled.

Response R2.10: The figure has been re-arranged accordingly.

Comment R2.11: In Fig. 5C, “IvyGap” is underlined in red. Fix this.

Response R2.11: The figure has been corrected.

Comment R2.12: Fig. 5E and 5F seem to be switched or mislabeled.

Response R2.12: The legend has been corrected to match the figure.

Comment R2.13: Please describe how the patients were divided into two groups.

Response R2.13: To address this we add text as follows (Page 8, Line 20-29), as follows:

“For this study, we used this 64 signature set to explore the functional landscape of GBM in a region-agnostic and unsupervised manner. For this analysis, we chose to focus on only CT and PAN, as these regions contained the highest proportion of pure tumor cells (nearly 100% by histology) compared to other regions like MVP, IT, and LE that contained a significant proportion of non-neoplastic cellular elements that could confound the analysis (See Supplementary Fig. 2-25). Hierarchical clustering of these purified tumour tissue regions (CT and PAN samples) (**Fig. 4a**) and t-distributed Stochastic Neighbour Embedding (tSNE) analysis (**Supplementary Fig. 29**) revealed the presence of two functional groups independently of the samples’ regional annotations. Moreover, this cluster structure also appeared to be independent of tumor origin with samples belonging to different niches from the same patient often segregating to different clusters (**Fig. 4a**).”

Comment R2.14: Enlarge font sizes for labels in Fig. 6D and other figures.

Response R2.14: the figure has been reformatted to improve its readability.

Comment R2.15: In page 7, line #10, “clinical outcomes of glioma patients from TCGA.” Is this glioma or GBM?

Response R2.15: We have now removed this data and the discussion related to survival benefits, as suggested by reviewer #3.

Comment R2.16: In page 7, line #10, “clinical outcomes of glioma patients from TCGA.” Is this glioma or GBM?

Response R2.16: Clinical outcomes have been removed from the main figures and discussion is limited to the text.

Comment R2.17: Software versions and database versions are missing in the Methods section.

Response R2.17: Software and databases used for mass spectrometry are referenced in the “statistical analysis” section of methods (Page 20, Lines 6-14) and we now include detailed description of packages and software used in “Selection informative gene expression signatures with RNA/protein concordance in GBM” section as follows:

New text included (Page 22, lines 16-19): “For our bioinformatic analysis throughout the study we used R 4.0.3 and packages GSVA 1.40.1, xgboost 1.4.1.1, ComplexHeatmap 2.8.0, psSubpathway 0.1.1, Rtsne 0.15 and modelTsn (https://github.com/oicr-gsi/modelTsn).”

Comment R2.18: Cannot find files or links for computer codes and R scripts used to generate the results.

Response R2.18: Page 8, line 9: We now provide link to the used R script.

New text included (Page 8, Line 16-19): We make this valuable set of XGBoost models to infer the level of activation of these 64 functional programs available for downloading (https://github.com/diamandis-lab/paper-prot-atlas-gbm) and via an online tool (https://cancerhub.shinyapps.io/prot-atlas-gbm/).

Reviewer 3:

This study represents an important resource for the community and the authors should be thanked for performing regionally-resolved proteomic analysis in GBM and making the data available to the community.

Comment R3.1: there are numerous survival analyses throughout the study and it is sometimes unclear what the message is; in the first part of the study, the analysis is focused on defining regionally-resolved protein-expression programs; but then typically only 1 protein is shown/chosen to correlate with survival; did other genes in e.g. PAN (other than AKAP12) also correlate with survival? (i.e. is it the abundance of PAN that correlated or is this specific to AKAP12); a similar question would be relevant for CD276 and MVP; is it MVP-globally or CD276-specific? It also seems that CD276 is quite highly expressed in PAN and it would be important to show the staining pattern in such areas as well?

Response R3.1: We agree with your comment that some niche-specific markers that correlate with other features of malignancy in gliomas (necrosis, MVP) may create confounders in the survival analysis and have removed them from the manuscript to avoid confusion. Regarding your comments of CD276 and PAN regions, while we did not observe PAN staining in our validation samples, we did find some CD276 staining within PAN areas from the human protein atlas resource and now include it in the new supplemental figure (Supplementary Fig. 27).

Comment R3.2: figure 5F: the survival analysis seems to not replicate so well in CPTAC dataset; would be important to explain discrepancies/again downplay clinical relevance if needed;

Response R3.2: Firstly, we apologize for the confusion due referencing the wrong figure panel (See R2.12) this has now been fixed. We confirm that the survival analysis does replicate within the CPTAC dataset as demonstrated by the included p-values of the statistical test (Fig. 5e-f).

Comment R3.3: the title/chosen nomenclature around KRAS and MYC is a somewhat simplification; other programs are part of those signatures (e.g. KRAS with mesenchymal and MYC with proneural); so one could also chose to use this "more established" GBM nomenclature; a rationale for focusing on KRAS and MYC naming/subprograms should be provided and the survival analyses should be done with (or compared to) the entire programs, not subprograms;

Response R3.3: We now provide a more detailed explanation behind our rationale for building a GBM disease model based on the expression status of the KRAS_TARGETS and MYC_TARGETS instead of using the GBM classification system proposed by Verhaak. Please see discussion to similar points raised by other reviews (See comment R1.5, R2.1). Overall, the ROC analysis (Fig. 4g) and the improved compactness of gene sets/modules of each axes was the main motivation behind our choice (Supplementary Fig. 31). As noted above this has important benefits when exploring GSEA of the proposed axes and critical differences in our pharmacogenomic analysis when compared to the more traditional GBM nomenclature. By having a simplified model that was not influenced by multiple expression “modules” with too many molecular dependencies that may obscure the most relevant biological differences, we believe we were able to nominate a number of subgroup-specific drug sensitivity differences, something less effectively done by traditional models. We thank all the reviewers for raising this important point as we believe the additional analysis suggested greatly improved the robustness of the chosen axes and the clinical implications.

Comment R3.4: In general, this reviewer feels that showing correlation to survival is ok, but many many markers have been suggested in GBM and tend to end up not being that useful; so a resource-focused paper, rather than too many survival curves would be preferable;

Response R3.4: We agree that individual markers rarely validate across multiple datasets in tumors like GBM with poor survival. We have removed the single protein survival curves. The KRAS specific survival curve, has been left in as it substantiated the aggressive and treatment resistant nature of these subpopulation of cells and is validated in both TCGA and CPTAC cohorts.

Minor Comments

Comment R3.5: colors in figure 1C are very hard to distinguish; a different color scheme should be used

Response R3.5: Thank you for your comment, we believe that you are referring to Fig. 2c as Fig. 1c is a schematic diagram. As such we have replaced the color scheme in Fig. 2c with one that is easier to view.

Comment R3.6: figure 3F is overstained

Response R3.6: We agree and now include images of an improved stain for AKAP12 in the same tumor (Fig. 3f).

REVIEWER COMMENTS

Reviewer #2 (Remarks to the Author):

In the revised manuscript, the authors improved their manuscript by clarifying the issues raised by this reviewer.

A specific detail:

1. Data portal website link below is not working properly.

http://the-brainatlas.herokuapp.com/dash/apps/app_gbm_atlas

Reviewer #3 (Remarks to the Author):

The authors have addressed all my concerns, and this represents a valid resource. For the publication, it is important to mention the size of the cohort as a limitation to the study.

Reviewer #4 (Remarks to the Author): Expert in machine learning, biostatistics, and proteogenomics

This manuscript presents an analysis of the GBM proteome focusing on profiling histomorphologic niches using laser capture microdissection followed by LC-MS. The study overlays transcription profiles over the proteomically characterized niches to develop a MYC- and KRAS-target signature that, when combined with hypoxia information, produces a model of intra-humoral heterogeneity that explains some of GBM's differential drug sensitivities and chemoresistance.

This review focuses on the statistical and machine learning data analysis methods used in the study. In general, the analysis performed in the study is sound, and analysis tools and algorithms are applied in a rational and logical manner. However, there are several comments and concerns that the authors need to address to ensure that the results are robust and clear:

* The authors state (pg 11, lines 23-24) that their exercise of using XGBoost models to infer CT niche specific signatures cannot be extended to proteomics data due to the incompatibility of their LFQ proteomics data with the CPTAC TMT data. The CPTAC TMT data includes MS1 precursor intensity for each peptide. "LFQ-equivalent" intensity for each channel can be derived by multiplying the MS1 precursor intensity by the TMT reporter ion fraction for that channel. Using this approach to replicate the analysis using proteome data would be very illustrative. In its current form, the MYC-KRAS axis is defined using the protein data, but all other application and validation is performed exclusively using RNA data.

* To characterize the immune cell compositions, the authors use CIBERSORT with the proteomics data as input. CIBERSORT is meant to be used with RNA expression data. It is unclear what the ramifications of this substitution are, and how robust the results are. At the very least, the authors should provide some comparison of CIBERSORT scores from RNA and protein (maybe for a different dataset) to show that the results are comparable, before using the protein-based CIBERSORT scores.

* XGBoost model for ssGSEA signatures: Use of XGBoost models to predict ssGSEA scores using protein and RNA expression data is unnecessary and a red-herring. There is no need to use a prediction model when the actual score can be calculated directly using ssGSEA. The authors' statement that this minimizes batch effects is not quite true: the decision trees underlying XGBoost are not translation and scale invariant, and any mitigation of batch effects is derived from appropriate normalization of the data.

* In the application of the XGBoost model for CT niche-specific inference, the authors state (pg 23, lines 14-15): "... sGSEA score from the original CT niche-specific sample divided by the CT content present in the synthetic sample." Is the CT content obtained from the concurrently trained predictor, or from the actual CT content based on the mixing ratio? How is this value determined when predicting scores for a new/test sample?

* In fig 5D, the authors show the CT-niche specific inference in TCGA-GBM data. Compared to the IvyGAP data, the negative slope in the TCGA-GBM data is much less pronounced, and the scatter is more spread out. One starts to wonder if the result is just a random occurrence. To address this concern, the authors should repeat the inference on a scrambled dataset and show that the pattern does not exist. Ideally, a permutation test could be used to determine the statistical significance of the slope in Fig 5D.

* The 64 signature set (pg 8 line 20) was derived in a region-agnostic manner. But, after the signatures have been identified, are there signatures that map to specific niches (ie how would the equivalent of Fig 2E look for the 64 signatures?)

* In the MVP and PAN compartments, the authors hypothesize deletion of memory CD4 T cells and no change in effector T cell/macrophage levels. These statements should be substantiated by an appropriate statistical test.

* It looks like all the figures and tables list nominal p-values, instead of p-values adjusted for multiple testing. The authors should use and report corrected p-values.

* When clustering the CT and PAN samples (Fig 4a), the authors need to perform a statistical enrichment test to formally check if cluster 1 or 2 is statistically enriched in CT or PAN samples.

* Supplementary Fig 34: What is "XGBoost interference", and how is it related to the WHOLE samples?

* Pg 11 line 20 should reference Fig 5f and line 21 should reference Fig 5e.

Point by Point response to reviewer comments

We thank all the reviewers for their efforts, time and useful feedback during the challenging times with COVID-19. Please find responses to your comments and suggestions below.

Reviewer # 2 (Remarks to the Author)

Comment R2.1: *In the revised manuscript, the authors improved their manuscript by clarifying the issues raised by this reviewer. A specific detail: Data portal website link below is not working properly: http://the-brainatlas.herokuapp.com/dash/apps/app_gbm_atlas*

Response R2.1: Thank you again for your feedback that improved our work. We have now also updated the broken data portal website link. See: <https://www.brainproteinatlas.org/dash/apps/GPA>

Reviewer #3 (Remarks to the Author):

Comment R3.1: *The authors have addressed all my concerns, and this represents a valid resource. For the publication, it is important to mention the size of the cohort as a limitation to the study.*

Response R3.1: Thank you again for your comments and effort in reviewing our work. We mention that the cohort size is a potential limitation in the revised manuscript (Page 15-16, Lines 31-2).

Reviewer #4 (Remarks to the Author): Expert in machine learning, biostatistics, and proteogenomics

Comment R4.1: *This manuscript presents an analysis of the GBM proteome focusing on profiling histomorphologic niches using laser capture microdissection followed by LC-MS. The study overlays transcription profiles over the proteomically characterized niches to develop a MYC- and KRAS-target signature that, when combined with hypoxia information, produces a model of intra-humoral heterogeneity that explains some of GBM's differential drug sensitivities and chemoresistance.*

This review focuses on the statistical and machine learning data analysis methods used in the study. In general, the analysis performed in the study is sound, and analysis tools and algorithms are applied in a rational and logical manner. However, there are several comments and concerns that the authors need to address to ensure that the results are robust and clear:

Response R4.1: Thank you for taking the time to review our work and your overall positive sentiment of our statistical/computational approaches. We highlight specific changes we made to the revised manuscript based on your comments and suggestions for improvement below.

Comment R4.2: *The authors state (pg 11, lines 23-24) that their exercise of using XGBoost models to infer CT niche specific signatures cannot be extended to proteomics data due to the incompatibility of their LFQ proteomics data with the CPTAC TMT data. The CPTAC TMT data includes MSI precursor intensity for each peptide. "LFQ-equivalent" intensity for each channel can be derived by multiplying the MSI precursor intensity by the TMT reporter ion fraction for that channel. Using this approach to replicate the analysis using proteome data would be very illustrative. In its current form, the MYC-KRAS axis is defined using the protein data, but all other application and validation is performed exclusively using RNA data.*

Response R4.2: Thank you for this suggestion to generate “LFQ-equivalent” data to further test our model in CTPAC. It is an intriguing method with the potential to bring LFQ and TMT datasets together. We conducted an exhaustive examination of the files deposited in the CPTAC data repositories, but were not able to find the MS1 precursor scores readily available for analysis. Even with MS1 values for the TMT data, we believe caution is needed for transforming TMT data into “LFQ-equivalent” values. We are unaware that reliable methods of converting MS data in such a manner has been formally demonstrated and accepted by the mass spectrometry community.

While the MYC-KRAS axis we describe was dependent on our newly generated protein dataset, it is important to emphasize it was defined in conjunction with the IVY GAP dataset and driven by gene sets found to have a high degree of spatial RNA/protein concordance (as indicated in the methods section). We consider this approach, that provides generalizability of our models to both transcriptional and proteomic datasets for validation and testing, a strength of our study.

Therefore, as an alternative analysis to further validate our MYC-KRAS axis and XGBoost model in CPTAC, we use available TMT protein and companion RNA values to show and confirm RNA/protein concordances of the relevant gene sets in CPTAC (KRAS, MYC, Hypoxia) (See newly generated **Supplementary Fig. 36**). Given this demonstrated concordance, we use the compatible CPTAC RNA data, to show that our CT-inference model again supports the existence of a MYC-KRAS axis in this cohort (See newly added **Figure panel 5e**).

Overall, as elaborated on in **Comment/Response 4.6**, we believe the observation of our MYC-KRAS axis in five independent datasets spanning spatially-defined, bulk, and single cell RNA and protein data provides strong experimental support for our proposed model.

***Comment R4.3:** To characterize the immune cell compositions, the authors use CIBERSORT with the proteomics data as input. CIBERSORT is meant to be used with RNA expression data. It is unclear what the ramifications of this substitution are, and how robust the results are. At the very least, the authors should provide some comparison of CIBERSORT scores from RNA and protein (maybe for a different dataset) to show that the results are comparable, before using the protein-based CIBERSORT scores.*

Response R4.3: This is a valid point regarding the use of protein data for CIBERSORT. We tried different datasets, as you suggested, and while there were some similarities, other immune cell type values did not show strong proteogenomic concordance using this tool. It is unclear if these differences are a limitation at the sample preparation level (e.g. different regions submitted for proteomics vs transcriptomics), the CIBERSORT algorithm or true proteogenomic differences in immune cell composition. We therefore decided to remove the CIBERSORT analysis given this limitation and that it was not critical for drawing the main conclusions of the study (defining and testing the presence of the triple axis).

We now support the speculative role of CD276 on local immunosuppression in glioblastoma by citing relevant immunohistochemically-based protein studies that have observed the relative paucity of tumor infiltrating lymphocytes in GBM and especially in niches containing microvascular proliferation and hypoxia (Page 7, Lines 20-23 and references 38-39).

Comment R4.4: *XGBoost model for ssGSEA signatures: Use of XGBoost models to predict ssGSEA scores using protein and RNA expression data is unnecessary and a red-herring. There is no need to use a prediction model when the actual score can be calculated directly using ssGSEA. The authors' statement that this minimizes batch effects is not quite true: the decision trees underlying XGBoost are not translation and scale invariant, and any mitigation of batch effects is derived from appropriate normalization of the data.*

Response: R4.4: We further elaborate on our motivation to use our approach for clarity and make appropriate changes as follows:

“Despite undeniable strengths of ssGSEA in estimating the activation status of biological programs, the use of machine learning is a very active area of research leading to development of new OMICs-based applications, such as inferences of drug susceptibility based on transcriptomic profiles. In this study, XGBoost served to learn molecular patterns of interest in a well-characterized dataset that could be later extrapolated to infer the functional status of samples from other datasets. The values inferred by the XGBoost models bear the context of the training sets, as these come scaled to a reference framework and less prone to interferences from non-target biological processes. Here, we therefore use ssGSEA as a tool to establish the ground truth across relevant datasets and apply machine learning tools to aid in extrapolation of the most relevant information to other datasets.”

We provide this additional discussion in the revised manuscript (See Page 16, Lines 7-16).

We agree with your later point, that perhaps the wordings “data normalization and batch effects” are non-contributory to this analysis. The previously included sentence (“... and therefore, their results are less influenced by batch effects”) has therefore been removed from the revised manuscript (See Page 22, Line 15).

Comment R4.5: *In the application of the XGBoost model for CT niche-specific inference, the authors state (pg 23, lines 14-15): “... sGSEA score from the original CT niche-specific sample divided by the CT content present in the synthetic sample.” Is the CT content obtained from the concurrently trained predictor, or from the actual CT content based on the mixing ratio? How is this value determined when predicting scores for a new/test sample?*

Response R4.5: For clarification, for CT-niche content inference model training, we used the known synthetic mixing ratios. These trained XGBoost models are then applied to test samples to inferring (predict) their relative niche composition (e.g. CT Content). This is described in the Methods section of the manuscript but to improve the clarity of our niche-specific ML approach, a schematic representation of the workflow has now been incorporated into the revised manuscript (See new **Supplemental Fig. 35**).

Comment R4.6: *In fig 5D, the authors show the CT-niche specific inference in TCGA-GBM data. Compared to the IvyGAP data, the negative slope in the TCGA-GBM data is much less pronounced, and the scatter is more spread out. One starts to wonder if the result is just a random occurrence. To address this concern, the authors should repeat the inference on a scrambled dataset and show that the pattern does not exist. Ideally, a permutation test could be used to determine the statistical significance of the slope in Fig 5D.*

Response R4.6: Thanks for your suggestion and comments. We now include p-values of the permuted linear model fitting in **Fig. 4c**, **Fig. 5c** and **Fig. 5d**. Furthermore, we now include additional analyses from CPTAC which again shows the expected inverse relationship of the MYC-KRAS axis (See **Fig 5e**) as we describe in response to comment R4.2. Overall, we universally observe this inverse trend between KRAS and MYC in our dataset (**Fig 4c**), IvyGAP (**Fig 5c**), single cell RNA data (**Fig 4h**) and bulk profiles from both TCGA (**Fig 5d**) and CPTAC datasets (**Fig 5e**). Together, we believe this provides strong experimental support for our triple-axis model.

We now discuss this finding as follows:

“Of note, we did expect to get a more subtle inverse relationship between the two axes in the TCGA (and CPTAC) datasets, as compared to IvyGAP, given that the data is derived from bulk samples. Even when inferring the make-up of the CT compartment, it is likely that CT-inferred patterns of the TCGA samples are derived from multiple and diverse CT-niches (combination of MYC and KRAS niches) dampening the more pronounced region-to-region differences we observed in both our dataset and that of the IvyGAP.” (Page 11, Lines 19-25).

***Comment R4.7:** The 64 signature set (pg 8 line 20) was derived in a region-agnostic manner. But, after the signatures have been identified, are there signatures that map to specific niches (ie how would the equivalent of Fig 2E look for the 64 signatures?)*

Response R4.7: As recommended, we now apply the 64 signatures across the entire sample cohort. The newly generated (**Supplementary Fig. 33**) illustrates the relative enrichment of each signature across all histomorphologic niches (mean ssGSEA scores).

***Comment R4.8:** In the MVP and PAN compartments, the authors hypothesize deletion of memory CD4 T cells and no change in effector T cell/macrophage levels. These statements should be substantiated by an appropriate statistical test.*

Response R4.8: We have now removed CIBERSORT analyses and associated sentence from the manuscript as discussed in Comment & Response 4.3.

***Comment R4.9:** It looks like all the figures and tables list nominal p-values, instead of p-values adjusted for multiple testing. The authors should use and report corrected p-values.*

Response R4.9: We have now provided corrected p-values that account for multiple testing. Please see relevant revised figures and tables.

***Comment R4.10:** When clustering the CT and PAN samples (Fig 4a), the authors need to perform a statistical enrichment test to formally check if cluster 1 or 2 is statistically enriched in CT or PAN samples.*

Response R4.10: Thank you for this suggestion. The lack of enrichment of CT or PAN within clusters 1 & 2 is now statistically substantiated using a chi-square test ($\chi^2 = 0.021$, $p = 0.886$). This statistical test of enrichment is now included in the revised manuscript on (Page 8, Lines 23-27) as follows:

“Page 8-Line 20: Hierarchical clustering of these purified tumour tissue regions (CT and PAN samples) (**Fig. 4A**) revealed the presence of two functional groups independent of the samples’ regional annotations, where these clusters lacked significant enrichment in CT and PAN samples, respectively ($X^2 = 0.021$, $p = 0.886$), and were further validated by t-distributed Stochastic Neighbour Embedding (tSNE) analysis (**Supplementary Fig. 28**).”

Comment R4.11: *Supplementary Fig 34: What is "XGBoost interference", and how is it related to the WHOLE samples?*

Response R4.11: We agree the XGBoost interference model is non-contributory to this figure based on the text referencing this figure. We have now replaced it with the ssGSEA values for consistency. The overall interpretation remains the same, specific niches (LE and MVP) can affect the expression level status of the relevant axes if disproportionally represented in specific samples. (Please see revised **Supplementary Fig. 34**).

Comment R4.12: *Pg 11 line 20 should reference Fig 5f and line 21 should reference Fig 5e.*

Response R4.12: Thank you for picking up this typo. It has now been corrected in the revised manuscript. (See Page 11, Lines 27-28).

REVIEWERS' COMMENTS

Reviewer #4 (Remarks to the Author):

The authors have done a thorough job of addressing almost all my comments and concerns. I just have one comment and one suggestion/question remaining:

* In Response R4.2, the authors state "We are unaware that reliable methods of converting MS data in such a manner has been formally demonstrated and accepted by the mass spectrometry community."

The method I proposed has been formally published in a recent CPTAC study (Cao et al., Cell 2021 on pancreatic ductal adenocarcinoma) in the "Proteomics and Phosphoproteomic Data Processing" Star*Methods section: "The ratios were converted back to abundances using the weighted sum of the MS1 intensities of the top three most intense peptide ions, with the weighting factor (computed for each PSM) taken as the ratio of the reference channel intensity to the summed reporter ion intensity (across all channels)."

Furthermore, if the the MS1 data for the GBM study was not included in the supplement, it should be available from the authors.

While using this LFQ-equivalent data approach would have been ideal, I find the authors' alternate approach of showing concordance across datasets acceptable.

* The authors should include a few sentences in the methods section on how they performed permutation testing (Response R4.6). I find it unusual for permutation p-values to be more extreme than nominal p-values. Given the "p-value < 2e-16" designation in Figures 5C-E, did the authors actually perform 10^{16} permutations? If not, I'm confused on how the permutation p-value was obtained.

Point by Point response to reviewer comments

Reviewer #4 (Remarks to the Author):

Comment R4.1: “The authors have done a thorough job of addressing almost all my comments and concerns. I just have one comment and one suggestion/question remaining”:

Response: Thank you once again for your time and efforts in reviewing and helping improve our work for publication. We address your remaining points in the revised manuscript as outlined below:

Comment R4.2 “In Response R4.2, the authors state “We are unaware that reliable methods of converting MS data in such a manner has been formally demonstrated and accepted by the mass spectrometry community.

The method I proposed has been formally published in a recent CPTAC study (Cao et al., Cell 2021 on pancreatic ductal adenocarcinoma) in the “Proteomics and Phosphoproteomic Data Processing” Star*Methods section: “The ratios were converted back to abundances using the weighted sum of the MS1 intensities of the top three most intense peptide ions, with the weighting factor (computed for each PSM) taken as the ratio of the reference channel intensity to the summed reporter ion intensity (across all channels). Furthermore, if the the MS1 data for the GBM study was not included in the supplement, it should be available from the authors.

While using this LFQ-equivalent data approach would have been ideal, I find the authors' alternate approach of showing concordance across datasets acceptable.”

Response R4.2: Firstly, we thank you for your approval in finding our alternative approach to addressing this comment acceptable. We now also mention the possibility of generating LFQ-equivalent values from TMT data to further the generalizability of our tools in future studies. We introduce this concept in the discussion section of the revised manuscript as outlined below.

Please see Page 16, Lines 10-15:

“While our niche-specific inference model was initially developed on LFQ proteomic data, a recent study highlighted how TMT datasets could be used to generate LFQ-equivalent values (Ref 48). This inference model, in theory, could therefore also be extended to other TMT-based datasets in future studies. This exciting prospect is especially supported by the comparison of ssGSEA proteogenomic concordances of relevant programs we highlighted in this study (e.g. KRAS/MYC/Hypoxia, Supplementary Fig. 36)”

Comment R4.3: “The authors should include a few sentences in the methods section on how they performed permutation testing (Response R4.6). I find it unusual for permutation p-values to be more extreme than nominal p-values. Given the “p-value < 2e-16” designation in Figures 5C-E, did the authors actually perform 10¹⁶ permutations? If not, I'm confused on how the permutation p-value was obtained.”

Response R4.3: Although the p-values of < 2e-16 from permuted linear model fits displayed on figures 5C and 5D were produced using methodologically sound statistical analyses (further discussion below), we acknowledge that these values can be easily misunderstood. Therefore, to avoid confusion, plots now include robust linear model fit p-values instead. These were specifically calculated using the `lm_robust` function of the `estimatr` R package to provide confirmatory null hypothesis testing (Figure 4C, Figure 5C and Figure 5D). The p-value of the permuted test is mentioned in the legend of Figure 4C only, given the proximity of the nominal p-value (p=0.027) to the level of significance. The included txt is as follows:

Legend of Figure 4C: Inverse correlation between the KRAS targets and MYC targets signatures in the protein dataset generated in this study (n=34 samples); additional null hypothesis testing was performed by permuted linear model fit analysis (lmPerm R package, p=0.024).

For clarification, permuted linear model fits were calculated using the lmp function of the lmPerm R package using the “Exact” method after 5,000 permutations. The reported values of $<2e-16$ correspond to a pre-determined lower limit to avoid unnecessarily long computations, and also, to prevent reporting p-values of “zero”. From the literature: “The statistical significance, as expressed in a p-value, is calculated as the fraction of permutation values that are at least as extreme as the original statistic, which was derived from non-permuted data”. Therefore, it is possible that the extreme nominal p-values noted in Figure 5 “shoot-up” to near-zero values when calculating the permuted linear models over 5,000 permutations.

The methods section now includes the following statement:

Page 21, lines 4-6: permuted linear model fit analysis was performed using the lmp function of the lmPerm R package with the “Exact” method and 5,000 permutations. Robust linear model fit analysis was carried out using the lm_robust function of the estimatr R package.